# Positive–Unlabeled Reinforcement Learning Distillation for On-Premise Small Models

**Zhiqiang Kou** [1 2]  **Junyang Chen** [1 2]  **Xin-Qiang Cai** [3]  **Xiaobo Xia** [4]  **Ming-Kun Xie** [3]
**Dong-Dong Wu** [5]  **Biao Liu** [1 2]  **Yuheng Jia** [1 2]  **Xin Geng** [1 2]  **Masashi Sugiyama** [3 5]  **Tat-Seng Chua** [4]

## Abstract

Due to constraints on privacy, cost, and latency, on-premise deployment of small models is increasingly common. However, most practical pipelines stop at supervised fine-tuning (SFT) and fail to reach the reinforcement-learning (RL) alignment stage. The main reason is that RL alignment typically requires either expensive human preference annotation or heavy reliance on high-quality reward models with large-scale API usage and ongoing engineering maintenance, both of which are ill-suited to on-premise settings. To bridge this gap, in this paper, we propose a positive–unlabeled (PU) RL distillation method for on-premise small-model deployment. Without human-labeled preferences or a reward model, our method distills the teacher's preference-optimization capability from black-box generations into a locally trainable student. For each prompt, we query the teacher once to obtain an anchor response, locally sample multiple student candidates, and perform anchor-conditioned self-ranking to induce pairwise or list-wise preferences, enabling a fully local training loop via direct preference optimization or group relative policy optimization. Theoretical analysis justifies that the induced preference signal by our method is order-consistent and concentrates on near-optimal candidates, supporting its stability for preference optimization. Experiments demonstrate that our method achieves consistently strong performance under a low-cost setting.

---

[1] School of Computer Science and Engineering, Southeast University, Nanjing, China [2] Key Laboratory of New Generation Artificial Intelligence Technology and Its Interdisciplinary Applications (Southeast University), Ministry of Education, China [3] RIKEN AIP [4] National University of Singapore [5] University of Tokyo . Correspondence to: Xiaobo Xia <xiaoboxia.uni@gmail.com>, Yuheng Jia <yhjia@seu.edu.cn>.

*Proceedings of the $43^{rd}$ International Conference on Machine Learning*, Seoul, South Korea. PMLR 306, 2026. Copyright 2026 by the author(s).

## 1. Introduction

Reinforcement learning (RL) has become a key mechanism for aligning large generative models, as preference-based paradigms such as reinforcement learning from human feedback (RLHF) (Ouyang et al., 2022), direct preference optimization (DPO) (Rafailov et al., 2023), and group relative policy optimization (GRPO) (Liu et al., 2024; Guo et al., 2025) can improve behaviors beyond static demonstrations. However, many real-world deployments prefer on-premise small expert models due to privacy, cost, latency, and compliance constraints (Touvron et al., 2023; Gunasekar et al., 2023; Shi et al., 2026a). In principle, such models follow the standard recipe of downstream supervised fine-tuning (SFT) adaptation followed by preference-based RL alignment (Zhou et al., 2023). Unfortunately, in practice, the RL stage is often infeasible on-premise: sustainable human preference labeling is costly, training and maintaining a reliable reward model is non-trivial, and high-quality alignment data are generally unavailable (Casper et al., 2023). While cloud providers may accumulate large-scale instruction and preference corpora, these data cannot be shared due to proprietary and regulatory constraints (Li et al., 2023a). As a result, on-premise expert models typically stop at SFT and lack RL-driven self-improvement capability.

As illustrated in Figure 1, it is natural to leverage a stronger foundation model to guide an on-premise small model. Nevertheless, existing solutions largely fall into two paradigms, both with clear limitations. The first paradigm is *imitation-based distillation* (Li et al., 2022; Lin et al., 2020; Shi et al., 2026b), shown in Figure 1(a), where the teacher provides responses (or logits) and the student is trained via SFT to mimic these demonstrations. While effective at transferring task competence, this paradigm relies on static examples and fails to impart preference-optimization–driven self-improvement. The second paradigm is *teacher-as-judge* RL (Yuan et al., 2024a; Lee et al., 2023), as shown in Figure 1(b). For each prompt, the student samples multiple candidate responses, and the teacher or an auxiliary reward model scores or ranks all candidates to produce scalar rewards for RL. Although this paradigm reduces explicit human annotation, it is cost-prohibitive and throughput-limited in on-premise

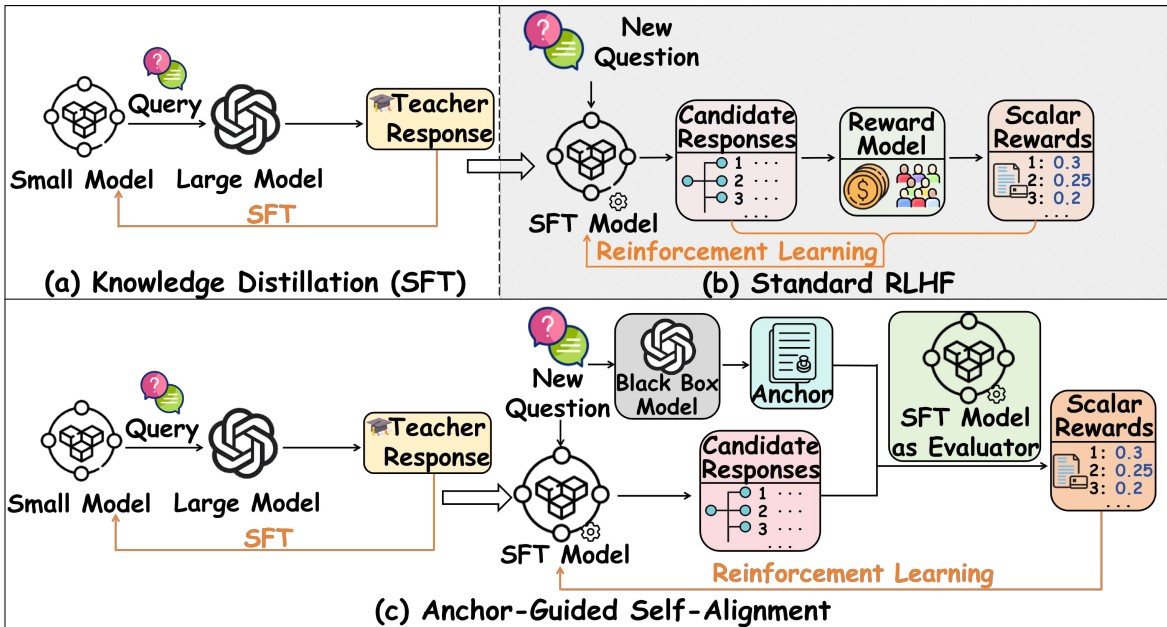

*Figure 1.* omparison among SFT, RLHF, and anchor-guided self-alignment (ours). For on-premise small-model deployment, training pipelines typically stop at the first-stage SFT: the small model imitates black-box teacher responses but fails to acquire preference-optimization–driven self-improvement. The second-stage RL alignment is hard to realize locally because it relies on large-scale human annotations or extensive reward-model calls. We propose a low-cost on-premise alignment strategy: for each query, we call the black-box teacher once to obtain an anchor and use anchor-guided local sampling and self-evaluation to generate scalar signals, enabling a practical SFT-to-RL loop under strict cost and deployment constraints.

settings. In particular, with $N$ prompts and $K$ candidates per prompt, the judge must evaluate approximately $NK$ responses, resulting in $\mathcal{O}(NK)$ teacher calls and largely increased token consumption and latency variance.

Based on the above observations, we raise a more fundamental question: can we distill the *preference-optimization* capability of RL from a black-box large model into an on-premise small model *without* human preference labels or a reward model? We formalize this as *Reinforcement Learning Capability Distillation* (RLCD), where the teacher model is only accessible via black-box generation while the student is white-box and can be locally sampled and updated. The goal is to enable *low-cost* and sustainable preference-based alignment under strict on-premise constraints.

To achieve RLCD, we adopt a *PU-based anchor preference induction* mechanism (Kiryo et al., 2017; Niu et al., 2016; Garg et al., 2021; Kou et al., 2025a;c). For each prompt x, we query the teacher once to obtain a high-quality response $y^+$ as an implicit *positive anchor*, and locally sample a set of student candidates $\{y_k\}_{k=1}^{K}$, treated as *unlabeled* data that may include both latent positives and inferior negatives. Conditioned on the anchor, the student performs *self-ranking* over its own candidates to induce relative preference relations, which are directly used by preference-optimization objectives such as DPO and GRPO. This forms a fully local "sampling–comparison–update" RL

loop. Crucially, the anchor is *not* a judge: the teacher is used only to produce the anchor and never scores or ranks student outputs. All preference signals are induced locally, reducing teacher dependence from $\mathcal{O}(NK)$ candidate evaluations to $\mathcal{O}(N)$ one-anchor-per-prompt queries and eliminating the need for a reward model, thereby meeting on-premise constraints on cost, throughput, and compliance.

We provide both theoretical and empirical support for our method. On the theoretical side, we show that the anchor-induced PU preference signal yields an order-consistent and well-concentrated listwise supervision, which justifies its use for stable preference optimization. Empirically, we evaluate our method on representative on-premise expert tasks across unimodal and multimodal settings, including writing, mathematical reasoning, and image captioning. Under the same or even lower teacher-query budget, our method consistently outperforms SFT, output distillation, and teacher-as-judge baselines. Together, these results demonstrate that, using only black-box teacher generations and without human preferences or a reward model, it is possible to distill the RL capability into small on-premise models, enabling a practical local alignment pipeline from SFT to RL. Our contributions are summarized as follows:

- We formulate RLCD, which studies how to distill preference-optimization–driven RL capability from a black-box large model into an on-premise small model

without human preferences or reward models.

- We propose two PU-based anchor preference induction methods that enable fully local alignment, which reduces teacher dependence effectively without requiring reward-model training.

- Theoretical guarantees are provided to show that the induced listwise preference supervision by our method can be order-consistent and concentrates on near-optimal candidates.

- Empirical results are offered to demonstrate consistent improvements over strong baselines across unimodal and multimodal on-premise tasks under the same or lower teacher-query budget.

## 2. Related Work

In this section, we review recent literature related to this work, including preference-based alignment, knowledge distillation (KD) for on-premise language models, and self-rewarded reinforcement learning.

### 2.1. Preference-Based Alignment

Reinforcement learning from human feedback (RLHF) has emerged as a dominant paradigm for aligning large language models with human intent and preferences (Ouyang et al., 2022). A canonical RLHF pipeline typically combines supervised fine-tuning (SFT) on curated instruction–response data with preference optimization based on human-annotated comparisons, often implemented through a learned reward model and RL algorithms such as proximal policy optimization (PPO) (Schulman et al., 2017) Subsequent studies have proposed more stable and simplified alternatives that directly optimize preference objectives (Rafailov et al., 2023), including reward-model-free formulations that bypass explicit reinforcement learning while still relying on preference pairs (Ethayarajh et al., 2024). Despite their effectiveness, most RLHF-style methods implicitly rely on the availability of at least one of the following: (i) curated instruction–answer datasets, (ii) human-annotated preference comparisons, or (iii) a reliable reward model. These assumptions are frequently violated in on-premise deployment settings, where privacy constraints and domain specificity restrict data sharing, and where constructing a high-quality reward model is costly, impractical, or even infeasible (Yuan et al., 2024b; Ye et al., 2025).

### 2.2. KD for On-Premise Language Models

Knowledge distillation (KD) (Gou et al., 2021; Park et al., 2019a; Wang & Yoon, 2021) transfers capabilities from a strong teacher model to a weaker student by using the teacher's outputs as supervision, and has been widely adopted to compress large models into deployable ones (Hinton et al., 2015; Kim & Rush, 2016; Zhang et al., 2019; Phuong & Lampert, 2019; Yang et al., 2023; Park et al., 2019b; Xiang et al., 2025; Yu et al., 2025; Shen et al., 2026b; Wei et al., 2025; Wu et al., 2025). In the context of large language models, a prevalent practice is instruction distillation, where a powerful teacher is queried to generate high-quality responses for a pool of prompts, and the student is subsequently trained via SFT to imitate these teacher-generated outputs (Sun et al., 2023; Wang et al., 2023; Hsieh et al., 2023; Li et al., 2023b; Shen et al., 2026a). Such strategies are particularly attractive for on-premise deployment, as they avoid direct exposure of private data and can operate with black-box access to hosted teacher models. However, most prior distillation methods primarily focus on transferring static knowledge or single-shot response quality, by matching the teacher's outputs on individual prompts (Ouyang et al., 2022). As a result, they largely overlook the distillation of the teacher's preference-optimization or self-improvement capabilities (Tunstall et al., 2023), which are typically acquired through alignment training.

### 2.3. Self-Rewarded Reinforcement Learning

A complementary line of research explored how language models can improve generation quality with minimal external supervision, e.g., through self-critique, self-refinement, or iterative bootstrapping (Madaan et al., 2023; Zelikman et al., 2022; Tian et al., 2024; Yin et al., 2025; Shridhar et al., 2024; He et al., 2025). Related works further attempted to replace human feedback with model-generated signals, including self-rewarding (Yuan et al., 2024a) and self-judging schemes (Lee et al., 2023) in which a model evaluates its own candidates and uses the resulting feedback for policy improvement (Yuan et al., 2024b; Bai et al., 2022; Zheng et al., 2023; Thawakar et al., 2025; Zhou et al., 2024). Despite their promise, purely self-driven signals are often unstable and prone to drift (Shumailov et al., 2023), whereas teacher-as-judge pipelines can be computationally expensive because they require scoring each candidate individually. In contrast, our method adopts a single teacher anchor as a high-confidence positive exemplar and elicits a groupwise soft preference distribution via in-context self-judging. This design enables iterative preference optimization using only local computation, without relying on explicit preference pairs or a learned reward model.

## 3. Method

In this section, we first offer some background knowledge and then introduce our method step by step. Finally, theoretical results are provided to support our claims.

## 3.1. Preliminaries

**Common on-premise alignment pipeline.** In an ideal on-premise deployment, one starts from a general base model $p_{\text{base}}(\cdot|\text{x})$ and performs supervised fine-tuning (SFT) on local SFT pairs $\mathcal{D}_{\text{sft}} = \{(\text{x}, \text{y})\}$ to obtain an instruction-following model $p_{\text{sft}}(\cdot|\text{x})$. Afterward, preference-based reinforcement learning (*e.g.*, RLHF or DPO) is applied using human-labeled preference tuples $\mathcal{D}_{\text{pref}} = \{(\text{x}, \text{y}^+, \text{y}^-)\}$ to produce the final aligned model $p_{\text{rl}}(\cdot|\text{x})$. Here, $\text{y}^+$ and $\text{y}^-$ denote the preferred and non-preferred responses for the same prompt x, respectively.

**Overall objective.** Given an unlabeled query pool $\mathcal{D}_{\text{x}}$ and black-box access to a teacher model $T$, our goal is to learn an on-premise student model $p_{\text{rl}}(\cdot|\text{x})$ that acquires reinforcement-learning capability under a strictly limited teacher-query budget. Specifically, we aim to: (i) query the teacher at most once per prompt to obtain an anchor response $\text{a} = T(\text{x})$; (ii) perform all subsequent exploration, comparison, and parameter updates locally using student-sampled candidates $U(\text{x}) = \{\text{y}_k\}_{k=1}^K$; and (iii) optimize the student via preference-based updates, without relying on human preference pairs or an explicit reward model. Below, we introduce the proposed method step by step. The algorithmic flow is provided in Algorithm 1.

## 3.2. Stage I: Teacher-Guided SFT

Although our ultimate goal is to distill reinforcement-learning capability, we first warm-start the on-premise student to ensure basic instruction following and a reasonable generation prior. This setting follows the standard cold-start practice adopted in prior work, *e.g.*, (Guo et al., 2025). Specifically, given an unlabeled query pool $\mathcal{D}_{\text{x}}$, we query the black-box teacher model $T$ once per prompt to obtain a response $\text{y} = T(\text{x})$ and construct an SFT dataset $\mathcal{D}_{\text{sft}} = \{(\text{x}, \text{y})|\text{x} \in \mathcal{D}_{\text{x}}\}$. We then fine-tune the on-premise student, starting from a base model $p_{\text{base}}(\cdot|\text{x})$ by minimizing the standard maximum-likelihood objective:

$$\min_{\theta} \mathbb{E}_{(\text{x},\text{y})\sim\mathcal{D}_{\text{sft}}}\big[-\log p_{\theta}(\text{y}|\text{x})\big]. \qquad (1)$$

This stage corresponds to conventional SFT using teacher-generated responses as training targets, without introducing any preference signals, ranking supervision, or reinforcement-learning updates. The resulting model, denoted as $p_{\text{sft}}(\cdot|\text{x})$, serves as a well-initialized starting point for Stage II, where the student is further improved through preference-based optimization rather than direct imitation.

## 3.3. Stage II: Reinforcement Learning Capability Distillation (RLCD)

**Anchor-conditioned PU self-evaluation.** Stage II adopts a positive-unlabeled (PU) formulation (Kiryo et al., 2017; Li et al., 2024; Niu et al., 2016; Li et al., 2026) tailored to low-cost on-premise alignment. For each query x, we associate each response with a latent quality indicator $\text{z} \in \{0, 1\}$, where $\text{z} = 1$ denotes a preferred response and $\text{z} = 0$ otherwise. As there are no ground-truth answers, human preference pairs, or reward model are available, the student's sampled responses are treated as unlabeled. We therefore query the black-box teacher once to obtain an anchor response a, which serves as a high-confidence *positive seed* as $T(\text{x}) = \text{a}$. The student then locally samples $K$ candidates $\{\text{y}_k\}_{k=1}^K$ with $\text{y}_k \sim p_{\theta}(\cdot|\text{x})$, forming an unlabeled set $U(\text{x}) = \{\text{y}_k\}_{k=1}^K$ that contain good and bad responses and may contain *latent positives*.

To transform the single positive seed into a dense training signal over $U(\text{x})$, we adopt *PU in-context inference*. The anchor a serves as an in-context reference for both correctness and style, enabling the student to *self-evaluate* its sampled candidates without external supervision. Specifically, for each $\text{y}_k \in U(\text{x})$, the student produces a scalar utility score $s_k = g_{\theta}(\text{x}, \text{a}, \text{y}_k)$, where $g_{\theta}$ is a scoring function induced by the SFT-initialized student model $p_{\text{sft}}(\cdot|\text{x})$, and only the relative magnitudes of $\{s_k\}_{k=1}^K$ will be used later. The resulting scores induce a listwise ranking over $U(\text{x})$ purely from local computation, without requiring an external judge or an explicit reward model.

A naive reranking that treats low-score candidates as negatives is unreliable under the PU setting, as $U(\text{x})$ may contain alternative correct responses. We therefore calibrate scores using the positive seed itself. Let $s^* = g_{\theta}(\text{x}, \text{a}, \text{a})$ and define the anchor-referenced margin $r_k := s_k - s^*$. We map this margin to a soft positivity confidence $u_k = \sigma(\gamma r_k)$, where $\gamma > 0$ is a softness parameter and $\sigma$ denotes the Sigmoid activation function. A PU-aware label distribution over the group is then constructed as follows:

$$w_k \propto u_k \exp(r_k/\tau), \quad D_{\text{x}}(k) = \frac{w_k}{\sum_{j=1}^K w_j} \in \Delta^{K-1}, \quad (2)$$

where $\tau$ is a temperature parameter and $\Delta$ denotes the probability simplex. This formulation allows a single teacher-provided anchor to induce a dense soft preference distribution $D_{\text{x}}$ over $U(\text{x})$ via PU-aware in-context self-evaluation. Given the induced soft preference distribution $D_{\text{x}} \in \Delta^{K-1}$ over $U(\text{x})$, we instantiate RLCD via label distribution learning (LDL) (Kou et al., 2024) and propose *LDL-GRPO*, which aligns the group-level policy allocation on $U(\text{x})$ with $D_{\text{x}}$ through distribution matching. We next introduce the *LDL-GRPO* formulation.

**LDL-GRPO.** Given the induced soft preference label distribution (Wu et al., 2026; Kou et al., 2025b) $D_{\text{x}} \in \Delta^{K-1}$ over the candidate set $\{\text{y}_k\}_{k=1}^K$, our goal is to update the on-premise student such that, within this group, it allocates probability mass in accordance with $D_{\text{x}}$. This realizes preference optimization *without* an external reward model: the

training signal is the distributional supervision $D_x$ derived from anchor-conditioned in-context self-evaluation. Since $D_x$ is defined on candidate indices, we first map the student policy to the same space by normalizing its *unnormalized sequence likelihoods* over the sampled group:

$$q_\theta(y_k|x) := \frac{\exp\big(\log p_\theta(y_k|x)\big)}{\sum_{j=1}^{K} \exp\big(\log p_\theta(y_j|x)\big)} = \frac{p_\theta(y_k|x)}{\sum_{j=1}^{K} p_\theta(y_j|x)}. \tag{3}$$

In practice, $q_\theta$ is computed via sequence log-likelihoods for numerical stability. We then minimize the divergence between the target label distribution $D_x$ and the candidate-normalized policy distribution $q_\theta(\cdot|x)$, while constraining the updated policy to stay close to a fixed reference model $p_{sft}$. The resulting *LDL-GRPO* objective is

$$\min_{\theta} \quad \mathbb{E}_x[\mathrm{KL}(D_x \| q_\theta(\cdot|x))] + \beta \, \mathbb{E}_x[\mathrm{KL}(p_\theta(\cdot|x) \| p_{sft}(\cdot|x))]. \tag{4}$$

where $\beta > 0$ is a hyperparameter to balance the two loss terms. Since $\mathrm{KL}(D_x\|q_\theta) = \sum_k D_x(k) \log D_x(k) - \sum_k D_x(k) \log q_\theta(y_k|x)$, the first term in Eq. (4) is equivalent, up to an additive constant independent of $\theta$, to minimizing the cross entropy $\sum_k D_x(k)\big(-\log q_\theta(y_k|x)\big)$. That is to say,

$$\min_{\theta} \quad \mathbb{E}_x \left[ -\sum_{k=1}^{K} D_x(k) \log p_\theta(y_k|x) + \log \sum_{j=1}^{K} p_\theta(y_j|x) \right] + \beta \, \mathbb{E}_x[\mathrm{KL}(p_\theta(\cdot|x) \| p_{sft}(\cdot|x))]. \tag{5}$$

This form makes clear that *LDL-GRPO* increases the likelihood of candidates weighted by $D_x$, while accounting for the group-wise normalization. Overall, *LDL-GRPO* implements a fully local loop: sample a candidate group → induce a PU-aware soft label distribution via anchor-conditioned self-evaluation → update the policy by matching its group-wise probability allocation to $D_x$, requiring neither human preference pairs nor a reward model.

### 3.4. Theoretical Justification

We provide the following two theoretical results to support our method. The first theorem ensures that $D_x$ is a strictly order-preserving listwise preference signal. The second theorem shows that $D_x$ concentrates on near-best candidates within the sampled set, with a gap controlled by two factors. These properties justify using $D_x$ as a label-distribution target for LDL-GRPO update, turning anchor-seeded PU preference induction into stable group-level supervision without explicit rewards or external judging.

**Theorem 3.1** (Order consistency). *For any* $i, j \in \{1, \ldots, K\}$, $r_i > r_j$ *implies* $D_x(i) > D_x(j)$.

*Proof sketch.* Let $f(r) = \log \sigma(\gamma r) + r/\tau$ so that $D_x(k) \propto \exp(f(r_k))$. Since $f'(r) = \gamma(1 - \sigma(\gamma r)) + 1/\tau > 0$ for

---

**Algorithm 1** Pseudo-Code for *LDL-GRPO*

1: **Input:** unlabeled prompt pool $\mathcal{D}_x$; black-box teacher $T$; student policy $p_\theta$ (initialized from $p_{sft}$); frozen reference $p_{sft}$; #candidates $K$; temperatures $\gamma, \tau$; KL weight $\beta$.
2: **Output:** aligned student $p_\theta$.
3: **while** not converged **do**
4:     Sample mini-batch $\{x_i\}_{i=1}^{B} \sim \mathcal{D}_x$.
5:     **for** $i = 1$ **to** $B$ **do**
6:         *One-shot teacher response:* $a_i \leftarrow T(x_i)$.
7:         *Local sampling:* draw $K$ candidates $y_{i,k} \sim p_\theta(\cdot|x_i)$.
8:         $s_i^* \leftarrow g_\theta(x_i, a_i, a_i)$.
9:         **for** $k = 1$ **to** $K$ **do**
10:            $r_{i,k} \leftarrow g_\theta(x_i, a_i, y_{i,k}) - s_i^*$.
11:            $\tilde{D}_i(k) \leftarrow \sigma(\gamma r_{i,k}) \exp(r_{i,k}/\tau)$.
12:            $\tilde{q}_i(k) \leftarrow p_\theta(y_{i,k}|x_i)$.
13:         **end for**
14:         $D_i \leftarrow \mathrm{Normalize}(\tilde{D}_i)$.
15:         $q_i \leftarrow \mathrm{Normalize}(\tilde{q}_i)$.
16:     **end for**
17:     Update $\theta$ by minimizing $\frac{1}{B} \sum_{i=1}^{B} \mathrm{KL}(D_i\|q_i) + \beta \frac{1}{B} \sum_{i=1}^{B} \mathrm{KL}(p_\theta(\cdot|x_i)\|p_{sft}(\cdot|x_i))$ (*cf.*, Eq. (4)).
18: **end while**

---

all $r$, $f$ is strictly increasing, which preserves the ordering induced by $\{r_k\}$. Detailed proof is provided in Appendix B.1. $\square$

**Remark 1.** Theorem 3.1 guarantees that the PU gating $\sigma(\gamma r)$ does not distort rankings: for any fixed $\gamma > 0$ and $\tau > 0$, $D_x$ induces exactly the same ordering as the calibrated margins $\{r_k\}_{k=1}^{K}$. In particular, replacing hard pair construction by listwise supervision with $D_x$ is consistent with the student's anchor-referenced comparisons.

**Theorem 3.2** (Near-optimality on the sampled set). *Let* $r_{\max} = \max_{1 \le k \le K} r_k$. *Then*

$$r_{\max} - \mathbb{E}_{k \sim D_x}[r_k] \le \tau \log K. \tag{6}$$

*Proof sketch.* Define the softmax distribution $\bar{D}(k) = \exp(r_k/\tau)/\sum_{j=1}^{K} \exp(r_j/\tau)$. A standard log-sum-exp bound gives $\mathbb{E}_{\bar{D}}[r_k] \ge r_{\max} - \tau \log K$ (Boyd & Vandenberghe, 2004). Moreover,

$$D_x(k) = \frac{\bar{D}(k)\, \sigma(\gamma r_k)}{\sum_{j=1}^{K} \bar{D}(j)\, \sigma(\gamma r_j)},$$

which is obtained by reweighting $\bar{D}$ with the increasing factor $\sigma(\gamma r_k)$, thereby shifting probability mass toward larger margins and ensuring $\mathbb{E}_{D_x}[r_k] \ge \mathbb{E}_{\bar{D}}[r_k]$. Combining the two inequalities yields (6). Detailed proof is provided in Appendix B.2. $\square$

**Remark 2.** The bound in (6) is controlled only by the sampling budget $K$ and temperature $\tau$: larger $K$ makes the sampled set more likely to contain strong candidates, while smaller $\tau$ tightens concentration around high-margin responses. Importantly, the PU term $\sigma(\gamma r)$ can only improve concentration relative to the pure softmax $\bar{D}$, since it monotonically upweights larger margins.

# 4. Experiments

## 4.1. Setups

**Datasets and Models.** We evaluate our method in unimodal and multimodal settings. For the unimodal setting, we consider two representative tasks: creative writing and mathematical reasoning. For creative writing, we employ WritingPrompts (Fan et al., 2018) and report results on two variants, WritingPrompts-CW and WritingPrompts-EU. For mathematical reasoning, we use Competition Math (Hendrycks et al., 2021) and evaluate on CompMath-Count and CompMath-Geometry. For the multimodal setting, we evaluate vision-language understanding on A-OKVQA (Schwenk et al., 2022) under two tasks: A-OKVQA-MC, a multiple-choice visual question answering task, and A-OKVQA-Rationale, which requires generating free-form rationales grounded in the image. Additional task details and evaluation prompts are provided in Appendix C.3.

**Backbone Models.** As on-premise student backbones, we exploit Qwen2.5-7B-Instruct (Yang et al., 2025b) and LLaMA3-8B-Instruct (Grattafiori et al., 2024) for unimodal tasks, and LLaVA-7B (Liu et al., 2023) and Qwen2.5-VL-7B-Instruct (Bai et al., 2025) for multimodal tasks. Model details and access information are provided in Appendix C.4.

**Evaluation Metric.** We conduct automatic A/B preference evaluation by comparing each baseline against GPT-4o (the 2024-11-20 version). Specifically, for each query, an external judge selects the better response according to task-specific criteria such as instruction following, correctness, and overall helpfulness. For unimodal text tasks, we adopt Qwen3-235B-A22B-Instruct (Yang et al., 2025a) as the judge model, while for multimodal vision-language tasks, we use Qwen3-VL-235B-A22B-Instruct (Yang et al., 2025a). We report both the raw win rate (Raw) and the length-controlled win rate (LC). The LC metric mitigates potential verbosity bias by enforcing comparable response lengths during evaluation. Ties are counted as half wins for both WR and LC. Additional details on the evaluation protocol and judge models are provided in Appendix C.1.

**Baselines.** We evaluate the following baselines that are strictly aligned with our stage-wise setting. Note that all methods share the same on-premise student backbone and the same stage-II prompts. They differ only in how supervision signals are constructed and how the student is updated.

(i). *SFT*. The on-premise model after the first-stage SFT only, with no second-stage preference optimization.

(ii). *SFT→SFT*. Starting from the SFT model, we query the black-box teacher on the stage II prompts and continue supervised fine-tuning on the teacher outputs, replacing RL with additional imitation learning.

(iii). *SinglePair-DPO* (Rafailov et al., 2023). Starting from the SFT model, for each stage II prompt, we obtain one teacher response and one student response, form a single preference pair where the teacher response is preferred, and optimize the student with direct preference optimization.

(iv). *Anchor-GRPO* (Guo et al., 2025). Starting from the SFT model, for each stage II prompt, we query the teacher once to obtain an anchor response, sample multiple student candidates locally, and use the on-premise SFT model as an anchor-conditioned evaluator to score these candidates. The resulting scalar (group-relative) rewards are then used to optimize the student via standard GRPO.

(v). *Self-PPO* (Schulman et al., 2017). Starting from the SFT model, we sample a single response per stage II prompt and obtain a scalar self-reward via on-premise self-evaluation. The student is then optimized with proximal policy optimization using this signal, without querying any external teacher or judge during training.

(vi). *AnchorRank-DPO* (Rafailov et al., 2023). Starting from the SFT model, we query the teacher once per stage-II prompt to obtain an anchor, sample multiple student candidates locally, and perform anchor-conditioned self-ranking to induce pairwise preferences. The student is then trained with DPO on these induced preferences.

**Implementation.** All methods (except for *SFT*) are implemented within a unified two-stage training pipeline to ensure fair and controlled comparison. We first perform SFT on teacher-generated responses to obtain a warm-started on-premise model that exhibits basic instruction-following ability. This SFT checkpoint is used to initialize all stage II variants. In stage II, all methods are trained on the same prompt set with identical decoding configurations and backbone architectures, and differ only in their optimization objectives and the construction of supervision signals. Specifically, we consider *SFT→SFT*, *SinglePair-DPO*, *Anchor-GRPO*, *Self-PPO*, *AnchorRank-DPO*, and *LDL-GRPO*. For anchor-guided methods, the black-box teacher is queried exactly once per prompt to obtain an anchor response, after which all candidate sampling, self-evaluation, and policy updates are performed entirely on-premise. This design ensures that performance differences stem from the learning objectives rather than from unequal teacher access or computational

*Table 1.* Main performance comparison of our methods and various baselines. The comparison covers six representative tasks spanning unimodal and multimodal domains. Within each case, the best result is highlighted in bold with a gray background, and the second-best result is underlined.

| Method | Qwen2.5-VL-7B | | | | LLaMA3-8B / LLaVA-7B | | | |
|---|---|---|---|---|---|---|---|---|
| | Raw ↑ | | LC ↑ | | Raw ↑ | | LC ↑ | |
| *Writing* | *Task 1: CFCW* | | | | *Task 2: PBFG* | | | |
| *SFT* | 0.538 | 0.452 | 0.573 | 0.513 | 0.568 | 0.487 | 0.482 | 0.372 |
| *SFT→SFT* | 0.352 | 0.337 | 0.372 | 0.327 | 0.462 | 0.387 | 0.417 | 0.312 |
| *SinglePair-DPO* | 0.492 | 0.447 | 0.477 | 0.437 | 0.533 | 0.432 | 0.683 | 0.482 |
| *Anchor-GRPO* | 0.508 | 0.452 | 0.548 | 0.487 | 0.573 | 0.513 | 0.673 | 0.508 |
| *Self-PPO* | 0.543 | 0.392 | 0.658 | 0.538 | 0.628 | 0.518 | 0.764 | 0.523 |
| *AnchorRank-DPO* | 0.573 | 0.417 | 0.663 | 0.573 | 0.643 | 0.472 | 0.814 | 0.608 |
| *LDL-GRPO* (ours) | **0.588** | **0.553** | **0.678** | **0.583** | **0.683** | **0.543** | **0.834** | **0.623** |
| *Math* | *Task 3: CountProb* | | | | *Task 4: Geometry* | | | |
| *SFT* | 0.533 | 0.487 | 0.382 | 0.427 | 0.613 | 0.583 | 0.457 | 0.437 |
| *SFT→SFT* | 0.558 | 0.533 | 0.442 | 0.417 | 0.678 | 0.658 | 0.608 | 0.588 |
| *SinglePair-DPO* | 0.462 | 0.452 | 0.392 | 0.382 | 0.457 | 0.442 | 0.397 | 0.382 |
| *Anchor-GRPO* | 0.558 | 0.538 | 0.422 | 0.397 | 0.628 | 0.608 | 0.452 | 0.432 |
| *Self-PPO* | 0.518 | 0.492 | 0.477 | 0.462 | 0.638 | 0.623 | 0.583 | 0.563 |
| *AnchorRank-DPO* | 0.538 | 0.492 | 0.492 | 0.482 | 0.633 | 0.598 | 0.633 | **0.618** |
| *LDL-GRPO* (ours) | **0.578** | **0.573** | **0.508** | **0.497** | **0.724** | **0.709** | **0.638** | 0.613 |
| *Multimodal* | *Task 5: A-OKVQA-MC* | | | | *Task 6: A-OKVQA-RG* | | | |
| *SFT* | 0.131 | 0.136 | 0.131 | 0.121 | 0.231 | 0.226 | 0.256 | 0.261 |
| *SFT→SFT* | 0.075 | 0.065 | 0.106 | 0.085 | 0.075 | 0.106 | 0.181 | 0.171 |
| *SinglePair-DPO* | 0.166 | 0.171 | 0.055 | 0.050 | 0.106 | 0.101 | 0.307 | 0.276 |
| *Anchor-GRPO* | 0.146 | 0.156 | 0.090 | 0.035 | 0.151 | 0.166 | 0.357 | 0.327 |
| *Self-PPO* | 0.206 | 0.216 | 0.136 | 0.126 | 0.241 | 0.231 | 0.382 | 0.362 |
| *AnchorRank-DPO* | 0.216 | 0.226 | 0.126 | 0.121 | 0.256 | 0.246 | 0.412 | 0.392 |
| *LDL-GRPO* (ours) | **0.236** | **0.231** | **0.146** | **0.141** | **0.261** | **0.256** | **0.432** | **0.417** |

budgets. Detailed implementation settings, training hyperparameters, and evaluation configurations are provided in Appendix C.

### 4.2. Results and Findings

We provide the comparison between our methods and baselines in Table 1. The results lead to the following findings.

**Finding 1.** Across all six tasks and backbones, our two anchor-based methods, *AnchorRank-DPO* and *LDL-GRPO*, consistently achieve the top-two performance under both Raw and LC metrics, outperforming all non-anchor baselines. This verifies the effectiveness of distilling preference optimization capability from a single teacher anchor with local exploration.

**Finding 2.** *LDL-GRPO* outperforms *AnchorRank-DPO* across tasks and model scales, indicating that label-distribution-based group supervision provides more robust and stable preference optimization than multi-pair DPO.

**Finding 3.** The advantages of anchor-based methods remain clear under LC evaluation, indicating that the improvements are not driven by verbosity. *LDL-GRPO* achieves the highest LC scores on all writing and multimodal tasks.

**Finding 4.** *SFT→SFT* underperforms *SFT*, while all preference-based methods yield clear gains, validating the necessity of a second-stage preference-optimization process.

**Finding 5.** *AnchorRank-DPO* consistently outperforms *SinglePair-DPO*, showing that inducing preferences from one anchor and multiple local candidates provides richer supervision than single-pair comparisons.

### 4.3. Ablation Study

We conduct detailed ablation studies across writing, mathematical reasoning, and multimodal tasks, with the comparative results of *SFT*, *Anchor-GRPO*, and our proposed *LDL-GRPO*, as summarized in Table 2. Experimental results indicate that *LDL-GRPO* achieves the highest Raw and

*Table 2.* Ablation study across six representative tasks. We compare *SFT*, *Anchor-GRPO*, and *LDL-GRPO*. Win rates (Raw) and length-controlled win rates (LC) are calculated against GPT-4o. In each case, the best result is bolded with gray shading, and the second-best result is underlined.

| Benchmark | Backbone | SFT | | Anchor-GRPO | | LDL-GRPO (ours) | |
|---|---|---|---|---|---|---|---|
| | | Raw ↑ | LC ↑ | Raw ↑ | LC ↑ | Raw ↑ | LC ↑ |
| *Domain I: Creative Writing* | | | | | | | |
| CFCW | Qwen2.5-7B | 0.538 | 0.452 | 0.508 | 0.452 | **0.588** | **0.553** |
| | LLaMA3-8B | 0.573 | 0.513 | 0.548 | 0.487 | **0.678** | **0.583** |
| PBFG | Qwen2.5-7B | 0.568 | 0.487 | 0.573 | 0.513 | **0.683** | **0.543** |
| | LLaMA3-8B | 0.482 | 0.372 | 0.673 | 0.508 | **0.834** | **0.623** |
| *Domain II: Mathematical Reasoning* | | | | | | | |
| CountProb | Qwen2.5-7B | 0.533 | 0.487 | 0.558 | 0.538 | **0.578** | **0.573** |
| | LLaMA3-8B | 0.427 | 0.382 | 0.422 | 0.397 | **0.508** | **0.497** |
| Geometry | Qwen2.5-7B | 0.613 | 0.583 | 0.628 | 0.608 | **0.724** | **0.709** |
| | LLaMA3-8B | 0.457 | 0.437 | 0.452 | 0.432 | **0.638** | **0.613** |
| *Domain III: Multimodal Understanding* | | | | | | | |
| MC | LLaVA-7B | 0.131 | 0.121 | 0.090 | 0.035 | **0.146** | **0.141** |
| | Qwen2.5-VL-7B | 0.131 | 0.136 | 0.146 | 0.156 | **0.236** | **0.231** |
| RG | LLaVA-7B | 0.256 | 0.261 | 0.357 | 0.327 | **0.432** | **0.417** |
| | Qwen2.5-VL-7B | 0.231 | 0.226 | 0.151 | 0.166 | **0.261** | **0.256** |

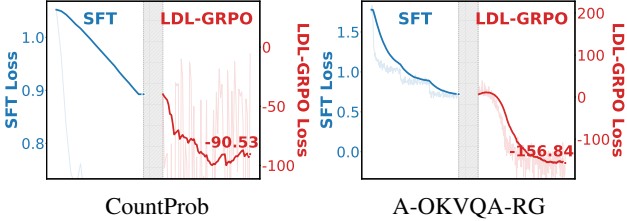

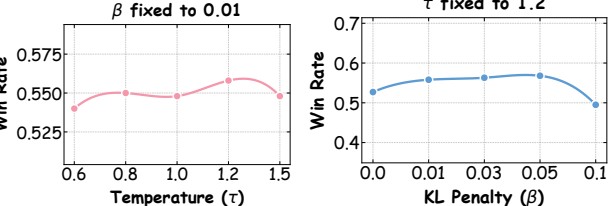

*Figure 2.* Two-stage convergence behavior from *SFT* to *LDL-GRPO*. The dashed vertical line marks the transition from *SFT* (Stage I) to *LDL-GRPO* (Stage II). Results are shown for Count-Prob with LLaMA3-8B and A-OKVQA-RG with LLaVA-1.5-7B. Across unimodal and multimodal tasks, *LDL-GRPO* exhibits stable loss trajectories after the transition, without sudden divergence.

*Figure 3.* Sensitivity analysis on CountProb (LLaMA3-8B). Representative sweeps with fixed $\beta$ (*left*) and fixed $\tau$ (*right*).

LC scores across all benchmarks and backbones, demonstrating stable performance gains over both the *SFT* baseline and *Anchor-GRPO*. Notably, *LDL-GRPO* exhibits superior optimization reliability compared to *Anchor-GRPO*. While *Anchor-GRPO* occasionally exhibits instability under noisy self-evaluation and in some cases underperforms the *SFT* baseline, such as in the A-OKVQA-MC task with the LLaVA-7B backbone, *LDL-GRPO* consistently delivers performance improvements. This behavior suggests that the label distribution learning mechanism effectively mitigates the adverse impact of evaluation noise. Furthermore, these improvements persist under LC evaluation, confirming that the gains arise from genuine enhancements in response quality rather than increased verbosity.

### 4.4. Convergence Analysis

Figure 2 illustrates the two-stage training dynamics from *SFT* to *LDL-GRPO*. After switching to *LDL-GRPO*, both unimodal (CountProb) and multimodal (A-OKVQA-RG) settings exhibit stable loss trajectories without abrupt oscillation or collapse. This indicates that anchor-conditioned and LDL-based group supervision enables a smooth transition from imitation learning to preference optimization, even in challenging multimodal reasoning tasks. We provided additional results of the convergence analysis in Appendix F.

### 4.5. Hyperparameter Sensitivity Analysis

As shown in Figure 3, our method is insensitive to moderate variations of the sampling temperature $\tau$ and the KL penalty $\beta$ on CountProb with LLaMA3-8B. With $\beta = 0.01$ fixed, sweeping $\tau \in [0.6, 1.5]$ yields a nearly flat win-rate

curve, indicating stable performance across a wide sampling regime. With $\tau = 1.2$ fixed, small-to-moderate $\beta$ values maintain comparable performance, whereas overly large $\beta$ noticeably degrades win rate, suggesting that excessive KL regularization restricts policy improvement. Based on this analysis, we adopt $\tau = 1.2$ and $\beta = 0.01$ in all experiments. Due to the limited space, more results of hyperparameter sensitivity analysis can be found in Appendix F.

## 5. Conclusion

In this work, we study a practical challenge that on-premise small expert models often stop at SFT and fail to benefit from a second-stage preference-optimization loop due to the high cost of human feedback, reward-model construction, and repeated teacher scoring. To address this, we propose an anchor-guided and fully local alignment strategy that queries a black-box teacher only once per prompt to obtain an anchor, and then relies on local multi-response sampling and self-evaluation to induce training signals for preference optimization. Building on this setup, we introduce *LDL-GRPO*, which converts anchor-induced comparisons into label-distribution-style group supervision and performs stable group-relative policy optimization. Across unimodal and multimodal benchmarks, the proposed method outperforms SFT-only and anchor-guided RL baselines under raw and length-controlled win rates, making SFT-to-RL practical for strict on-premise deployment.

## 6. Limitation

While our method enables a fully local and low-cost alignment pipeline under strict on-premise constraints, it still relies on several design choices that need further investigation. First, the quality of the induced preference signal depends on the student's anchor-conditioned self-evaluation, which may be imperfect when the student is still relatively weak or when the task requires fine-grained judgment. Although the proposed PU formulation and label-distribution learning help mitigate noise and instability, exploring more robust self-evaluation mechanisms remains an interesting direction. Second, our current implementation requires sampling multiple candidates per prompt to construct group-level supervision, which introduces additional computation compared to single-response updates. Improving the efficiency of candidate generation and reuse, or reducing the required group size without hurting stability, is left for future work.

## Impact Statement

This work focuses on enabling practical preference-based alignment for on-premise small models under strict constraints on privacy, cost, and deployment. By reducing reliance on human preference annotation, reward-model training, and repeated teacher scoring, the proposed method can lower the barrier to deploying aligned models in resource-constrained or privacy-sensitive environments, such as enterprise systems, edge devices, and regulated domains. In this sense, the work could contribute to more accessible and sustainable alignment pipelines beyond large-scale cloud settings. At the same time, our method relies on self-induced preference signals guided by a teacher-provided anchor, which may reflect the biases or limitations of both the teacher and the student models. As with other alignment and distillation techniques, care should be taken when deploying the resulting models in high-stakes applications, and additional safeguards or human oversight may still be required. We hope this work encourages further research on low-cost, transparent, and reliable alignment methods that balance practical feasibility with responsible deployment.

## Acknowledgements

This work was supported by the Jiangsu Science Foundation under Grants BG2024036 and BK20243012, the National Natural Science Foundation of China under Grants 62125602, U24A20324, 92464301, U24A20322, and 62576094, the New Cornerstone Science Foundation through the XPLORER PRIZE, the Fundamental Research Funds for the Central Universities under Grant 2242025K30024, the Big Data Computing Center of Southeast University, JST ASPIRE under Grant JPMJAP2405, and JSPS KAKENHI under Grant JP26K21323.

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

# Appendix Table of Contents

# A. Notation and Definition

*Table A.3.* Mathematical notations and definitions

| Symbol | Definition |
|---|---|
| x | A prompt or query (scalar or structured input, e.g., text or multimodal input). |
| y | A response sequence generated for prompt x. |
| $\mathcal{D}_{\mathrm{x}}$ | Unlabeled on-premise prompt pool. |
| $T$ | Black-box teacher model, accessed only via generation. |
| $\mathrm{a} = T(\mathrm{x})$ | Teacher-generated response for prompt x, used as an anchor. |
| $p_{\mathrm{base}}(\cdot|\mathrm{x})$ | Base student policy before supervised fine-tuning. |
| $p_{\mathrm{sft}}(\cdot|\mathrm{x})$ | Student policy after supervised fine-tuning (Stage I). |
| $p_{\theta}(\cdot|\mathrm{x})$ | Trainable student policy in Stage II, parameterized by $\theta$. |
| $p_{\mathrm{ref}}(\cdot|\mathrm{x})$ | Frozen reference policy for KL regularization (set to $p_{\mathrm{sft}}$). |
| $U(\mathrm{x}) = \{\mathrm{y}_k\}_{k=1}^{K}$ | Candidate response set sampled from the student for prompt x. |
| $K$ | Number of sampled candidate responses per prompt. |
| $g_{\theta}(\mathrm{x}, \mathrm{a}, \mathrm{y})$ | Anchor-conditioned self-evaluation score produced by the student. |
| $s^* = g_{\theta}(\mathrm{x}, \mathrm{a}, \mathrm{a})$ | Anchor self-score used as a calibration baseline. |
| $r_k$ | Anchor-referenced margin for candidate $\mathrm{y}_k$, defined as $r_k = s_k - s^*$. |
| $\sigma(\cdot)$ | Sigmoid function used for PU confidence calibration. |
| $\gamma$ | Sharpness parameter controlling PU confidence transition. |
| $\tau$ | Temperature parameter controlling the concentration of softmax weighting. |
| $D_{\mathrm{x}} \in \Delta^{K-1}$ | PU-induced soft preference label distribution over $U(\mathrm{x})$. |
| $q_{\theta}(\mathrm{y}_k|\mathrm{x})$ | Candidate-normalized student policy probability over $U(\mathrm{x})$. |
| $\beta$ | Weight of KL regularization against the reference policy. |

# B. Theoretical Proofs

## B.1. Proof of Theorem 3.1

*Proof.* Define the function

$$f(r) := \log \sigma(\gamma r) + \frac{r}{\tau}.$$

Then the induced distribution can be written as

$$D_{\mathrm{x}}(k) \propto \exp\big(f(r_k)\big).$$

We first show that $f$ is strictly increasing. Taking the derivative,

$$f'(r) = \frac{d}{dr} \log \sigma(\gamma r) + \frac{1}{\tau} = \gamma\big(1 - \sigma(\gamma r)\big) + \frac{1}{\tau}.$$

Since $\gamma > 0$, $\tau > 0$, and $0 < \sigma(\gamma r) < 1$, we have

$$f'(r) > 0 \quad \text{for all } r \in \mathbb{R}.$$

Hence $f$ is strictly increasing.

Therefore, for any $i, j$,

$$r_i > r_j \implies f(r_i) > f(r_j) \implies \exp(f(r_i)) > \exp(f(r_j)) \implies D_{\mathrm{x}}(i) > D_{\mathrm{x}}(j).$$

This proves that $D_{\mathrm{x}}$ strictly preserves the ordering induced by the margins $\{r_k\}$. $\qquad\square$

## B.2. Proof of Theorem 3.2

*Proof.* We first define the standard softmax distribution

$$\bar{D}(k) := \frac{\exp(r_k/\tau)}{\sum_{j=1}^{K} \exp(r_j/\tau)},$$

and define the normalization constant

$$Z := \sum_{j=1}^{K} \exp(r_j/\tau).$$

We then relate $\log Z$ to the expected margin and the entropy of $\bar{D}$. By definition,

$$\log \bar{D}(k) = \frac{r_k}{\tau} - \log Z.$$

Therefore, the Shannon entropy of $\bar{D}$ satisfies

$$\begin{aligned}
H(\bar{D}) &:= - \sum_{k=1}^{K} \bar{D}(k) \log \bar{D}(k) \\
&= - \sum_{k=1}^{K} \bar{D}(k) \left( \frac{r_k}{\tau} - \log Z \right) \\
&= - \frac{1}{\tau} \sum_{k=1}^{K} \bar{D}(k) r_k + \log Z \sum_{k=1}^{K} \bar{D}(k).
\end{aligned}$$

Since $\sum_{k=1}^{K} \bar{D}(k) = 1$, we obtain

$$\log Z = \frac{1}{\tau} \mathbb{E}_{\bar{D}}[r_k] + H(\bar{D}),$$

which gives

$$\log \sum_{k=1}^{K} \exp(r_k/\tau) = \frac{1}{\tau}\mathbb{E}_{\bar{D}}[r_k] + H(\bar{D}).$$

Next, we bound the two terms. On the one hand, the entropy is upper-bounded by

$$H(\bar{D}) \leq \log K.$$

On the other hand,

$$\log \sum_{k=1}^{K} \exp(r_k/\tau) \geq \log \exp(r_{\max}/\tau) = \frac{r_{\max}}{\tau}.$$

Combining the above inequalities yields

$$\mathbb{E}_{\bar{D}}[r_k] \geq r_{\max} - \tau \log K.$$

Observe that $D_{\mathrm{x}}$ can be written as

$$D_{\mathrm{x}}(k) = \frac{\bar{D}(k)\,\sigma(\gamma r_k)}{\sum_{j=1}^{K} \bar{D}(j)\,\sigma(\gamma r_j)}.$$

Since $\sigma(\gamma r)$ is a strictly increasing function of $r$, multiplying $\bar{D}$ by $\sigma(\gamma r_k)$ shifts probability mass toward larger margins. Formally, for the random variable $R$ taking values $\{r_k\}$ under $\bar{D}$, the covariance $\mathrm{Cov}_{\bar{D}}(R, \sigma(\gamma R)) \geq 0$. Thus,

$$\mathbb{E}_{k \sim D_{\mathrm{x}}}[r_k] \geq \mathbb{E}_{k \sim \bar{D}}[r_k].$$

Combining the two steps yields

$$r_{\max} - \mathbb{E}_{k \sim D_{\mathrm{x}}}[r_k] \leq r_{\max} - \mathbb{E}_{k \sim \bar{D}}[r_k] \leq \tau \log K.$$

This completes the proof. $\square$

# C. Supplementary Experimental Instructions

## C.1. Evaluation Protocol and Judge Models

**Judge Models.** For automatic preference evaluation, we use large instruction-tuned foundation models as external judges. Specifically, for unimodal text-only tasks, we adopt Qwen3-235B-A22B-Instruct, available at `https://huggingface.co/Qwen/Qwen3-235B-A22B-Instruct-2507`. For multimodal tasks, we use Qwen3-VL-235B-A22B-Instruct, available at `https://huggingface.co/Qwen/Qwen3-VL-235B-A22B-Instruct`. The judge prompt used for automatic evaluation is shown below.

```
 You will judge two responses to the same question.
Choose the better one overall (correctness, grounding, clarity, completeness).
You are given the reference (ground-truth answer) to help judge correctness.
Reply with exactly one character:
A = Response A is better
B = Response B is better
T = Tie / too close to call

=== Question / Prompt ===
{task_prompt}

=== Reference (Ground Truth) ===
{gt}

=== Response A ===
{ans_model}

=== Response B ===
{ans_gpt}
```

*Figure C.4.* Prompt used for automatic pairwise preference evaluation.

## C.2. Teacher Query Budget and Cost Analysis

We summarize the teacher-query complexity of different alignment strategies under a unified setting with $N$ prompts and $K$ candidates per prompt.

*Table C.4.* Teacher-query complexity comparison.

| Method | Teacher Calls Order |
|---|:---:|
| *SFT* | $N$ |
| *SinglePair-DPO* | $N$ |
| *Teacher-as-Judge RL* | $NK$ |
| *AnchorRank-DPO* | $N$ |
| *LDL-GRPO* | $N$ |

Both *AnchorRank-DPO* and *LDL-GRPO* require only one black-box teacher generation per prompt, while all candidate scoring and policy updates are performed fully on-premise.

## C.3. Datasets and Evaluation Tasks

**WritingPrompts.** We use the WritingPrompts dataset (Fan et al., 2018) for creative writing evaluation. The original dataset is available at `https://arxiv.org/abs/1805.04833`. We construct two variants: (i) WritingPrompts-CW, focusing on constraint-following creative writing, and (ii) WritingPrompts-EU, emphasizing expressive and stylistic diversity. Table C.5 provides representative examples from the WritingPrompts dataset used in our evaluation, including the input prompt and model-generated responses.

*Table C.5.* Representative examples from the WritingPrompts dataset.

| Field | Content |
|---|---|
| **Prompt (Case 1)** | *[EU]* As a treat to fans, George R. R. Martin writes a character inspired by himself into the final book of the *A Song of Ice and Fire* series. That character suffers the most gruesome fate of all. |
| **Reference** | A dramatic courtroom-style scene in which the author-character is judged and sentenced, written with dark humor and theatrical dialogue. |
| **Model Output** | A narrative depicting the author-character pleading innocence before being condemned to a brutal fate, featuring exaggerated dialogue, irony, and grim imagery consistent with the prompt. |
| **Prompt (Case 2)** | *[CW]* Each character can only speak one sentence. |
| **Reference** | A tense short story set aboard a damaged submarine, where each character's single-sentence dialogue advances the plot toward a sacrificial ending. |
| **Model Output** | A compact narrative obeying the one-sentence-per-character constraint, using internal monologue and sparse dialogue to convey urgency and moral tension. |

*Table C.6.* Representative examples from the Competition Math dataset used in our evaluation.

| Subset | Prompt | Ground Truth Answer |
|---|---|---|
| **CompMath-Count** | Solve the following competition math problem: Bob's password consists of a non-negative single-digit number followed by a letter and another non-negative single-digit number (which could be the same as the first one). What is the probability that Bob's password consists of an odd single-digit number, followed by a letter, and a positive single-digit number? | Exactly 5 out of the 10 non-negative single-digit numbers are odd, giving probability $\frac{1}{2}$. The second character is always a letter. For the last character, 9 out of 10 digits are positive. Thus, the desired probability is $\frac{1}{2} \times 1 \times \frac{9}{10} = \boxed{\frac{9}{20}}$. |
| **CompMath-Geometry** | Solve the following competition math problem: A wire is cut into two pieces, one of length $a$ and the other of length $b$. The piece of length $a$ is bent to form an equilateral triangle, and the piece of length $b$ is bent to form a regular hexagon. If the triangle and the hexagon have equal area, what is $\frac{a}{b}$? | The triangle side length is $\frac{a}{3}$ and its area is $\frac{\sqrt{3}}{4}(\frac{a}{3})^2 = \frac{a^2\sqrt{3}}{36}$. The hexagon side length is $\frac{b}{6}$ and its area is $\frac{3\sqrt{3}}{2}(\frac{b}{6})^2 = \frac{b^2\sqrt{3}}{24}$. Equating the two areas yields $\frac{a^2}{b^2} = \frac{3}{2}$, hence $\frac{a}{b} = \boxed{\frac{\sqrt{6}}{2}}$. |

**Competition Math.** We evaluate mathematical reasoning using the Competition Math dataset (Hendrycks et al., 2021), available at https://arxiv.org/abs/2103.03874. We report results on two subsets: CompMath-Count, which emphasizes counting and arithmetic reasoning, and CompMath-Geometry, which focuses on geometric problem solving. Table C.6 provides representative examples from the Competition Math dataset used in our evaluation, including the input prompt and model-generated responses.

**A-OKVQA.** For multimodal evaluation, we use the A-OKVQA benchmark (Schwenk et al., 2022), available at https://arxiv.org/abs/2206.01718. We consider two tasks: A-OKVQA-MC, a multiple-choice VQA task, and A-OKVQA-Rationale, which requires generating free-form rationales grounded in the image content. Table C.7 provides representative examples from the A-OKVQA dataset used in our evaluation, including the input prompt and model-generated responses.

### C.4. Backbone Models and Implementation Details

For unimodal on-premise deployment, we use Qwen2.5-7B-Instruct (https://huggingface.co/Qwen/Qwen2.5-7B-Instruct) and LLaMA3-8B-Instruct (https://huggingface.co/meta-llama/Meta-Llama-3-8B-Instruct).

*Table C.7.* Example cases from the A-OKVQA benchmark used in our evaluation.

| Task | Prompt (Image + Question) | Ground Truth Output |
|------|---------------------------|---------------------|
| A-OKVQA-MC | **Question:** What color is reflected strongly off the metal cabinet cases? 
 **Options:** 
 (0) purple 
 (1) red 
 (2) blue 
 (3) yellow | **Answer:** red |
| A-OKVQA-Rationale | **Question:** What does he use to build momentum? 
 (The input includes an image of a person skateboarding.) | **Rationale:** 
 A guy is on a skateboard as he cruises down the street. He uses the bottom part of his leg to push against the ground and propel himself forward. |

For multimodal tasks, we use LLaVA-7B (https://github.com/haotian-liu/LLaVA) and Qwen2.5-VL-7B-Instruct (https://huggingface.co/Qwen/Qwen2.5-VL-7B-Instruct).

All models are deployed locally without external API calls during alignment and evaluation.

**Anchor-conditioned self-evaluation (scalar scoring).** For each query x, an anchor response a generated by the teacher, and a candidate response y sampled from the student, we prompt the student to perform anchor-conditioned self-evaluation as follows:

> **Question:**
> A jar contains 7 red marbles and 5 blue marbles. Two marbles are drawn without replacement. What is the probability that both marbles are red?
>
> **Reference Answer:**
> There are $\binom{12}{2}$ ways to draw 2 marbles. Favorable outcomes are drawing 2 from the 7 red marbles: $\binom{7}{2}$. Thus the probability is $\frac{\binom{7}{2}}{\binom{12}{2}} = \frac{21}{66} = \frac{7}{22}$.
>
> **Candidate Answer:**
> The probability is $(7/12) \cdot (6/11) = 42/132 = 7/22$.
>
> **Scalar Score (example output):** 0.95

**Anchor self-calibration.** To calibrate scores under the positive–unlabeled setting, we also evaluate the anchor against itself using the same prompt structure. Specifically, we set the candidate answer to be identical to the anchor:

> **Question:**
> A fair coin is flipped 4 times. What is the probability of getting exactly 3 heads?
>
> **Reference Answer:**
> There are $\binom{4}{3} = 4$ sequences with exactly 3 heads, and $2^4 = 16$ total outcomes, so the probability is $4/16 = 1/4$.
>
> **Candidate Answer:**
> There are $\binom{4}{3} = 4$ sequences with exactly 3 heads, and $2^4 = 16$ total outcomes, so the probability is $4/16 = 1/4$.
>
> **Scalar Score (example output):** 1.00

# D. Reproducibility

We provide implementation details, involving illustrative algorithm descriptions in Section 3, Section 4, and Appendix C, and pseudo-code in Algorithm 1. The code will be publicly released for reproducibility.

## E. Use of LLMs in Writing

We used a large language model (LLM) solely to polish the writing and correct grammatical issues during the preparation of this paper. The LLM was not involved in idea generation, experiment design, or analysis. All scientific contributions are entirely made by the authors.

## F. Supplementary Results

**Two-stage convergence from SFT to LDL-GRPO.** Figures F.9 and F.10 visualize the training dynamics of our two-stage pipeline on representative model–task pairs. In each panel, the *SFT stage* (blue; left y-axis) exhibits a smooth decrease of the supervised objective, indicating stable imitation learning. After switching to *LDL-GRPO* (red; right y-axis; gray band marks the transition), the *LDL-GRPO* objective consistently drops and then stabilizes, demonstrating that anchor-induced group supervision yields a well-behaved preference-optimization signal. Importantly, we observe no abrupt divergence or oscillation across both unimodal writing tasks (*e.g.*, CountProb, Geometry, CFCW, PBFG) and multimodal benchmarks (*e.g.*, MC, RG), and the same stable trend holds for different backbones (*e.g.*, LLaMA3-8B, Qwen2.5-7B, LLaVA/Qwen2.5-VL).

**Compact sensitivity plots.** Figures F.11 and F.12 provide compact sweeps over the sampling temperature $\tau$ and KL penalty $\beta$ on CountProb and A-OKVQA-RG, respectively. Across fixed-$\beta$ sweeps, the win rate remains largely flat over a wide range of $\tau$, suggesting that our anchor-guided training is not overly sensitive to sampling stochasticity. Across fixed-$\tau$ sweeps, small to moderate $\beta$ values achieve comparable performance, whereas overly large $\beta$ tends to reduce win rate, consistent with over-regularization that restricts policy improvement. These results support that LDL-GRPO has a broadly stable operating region for hyperparameters.

**Visualization of experimental results.** We provide the visualization of experimental results, which is achieved by our method. Details can be checked in Table F.8, Figure F.5, Figure F.6, Figure F.7, Figure F.8.

*Table F.8.* Full evaluation results on the Competition Math dataset.

| Dataset | Prompt | Ground Truth | Pred (Ours) | Pred (GPT-4o) |
|---|---|---|---|---|
| **Counting-Probability** | Solve the following competition math problem: How many natural-number factors does $N$ have if $N = 2^4 \cdot 3^3 \cdot 5^2 \cdot 7^2$? | Any positive integer divisor of $N$ must take the form $2^a \cdot 3^b \cdot 5^c \cdot 7^d$ where $0 \leq a \leq 4$, $0 \leq b \leq 3$, $0 \leq c \leq 2$, $0 \leq d \leq 2$. In other words, there are 5 choices for $a$, 4 choices for $b$, 3 choices for $c$, and 3 choices for $d$. So there are $5 \cdot 4 \cdot 3 \cdot 3 = \boxed{180}$ natural-number factors of $N$. | We are given the prime factorization of a natural number $N$: $N = 2^4 \cdot 3^3 \cdot 5^2 \cdot 7^2$ We are asked to find how many **natural-number factors** $N$ has.

**Step 1: Use the formula for the number of divisors**
For a number $N$ with prime factorization: $N = p_1^{e_1} \cdot p_2^{e_2} \cdot \ldots \cdot p_k^{e_k}$ The number of **divisors** (including 1 and $N$ itself) is: $(e_1+1)(e_2+1)\cdots(e_k+1)$

**Step 2: Apply the formula to our $N$**
From the factorization:
• $2^4 \rightarrow$ exponent $4 \rightarrow$ contributes $4+1=5$
• $3^3 \rightarrow$ exponent $3 \rightarrow$ contributes $3+1=4$
• $5^2 \rightarrow$ exponent $2 \rightarrow$ contributes $2+1=3$
• $7^2 \rightarrow$ exponent $2 \rightarrow$ contributes $2+1=3$
Multiply these together: $(4+1)(3+1)(2+1)(2+1) = 5\cdot4\cdot3\cdot3$ Now compute: $5 \cdot 4 = 20$ $20 \cdot 3 = 60$ $60 \cdot 3 = 180$

**Final Answer:** $\boxed{180}$

✓**Check:**
Let's double-check by writing out the full product: $5 \cdot 4 \cdot 3 \cdot 3 = 5 \cdot 4 = 20$ $20 \cdot 3 = 60$ $60 \cdot 3 = 180$ Yes, matches. | To determine the number of natural-number factors of $N$, we use the formula for the total number of factors of a number expressed in its prime factorization form.

Given: $N = 2^4 \cdot 3^3 \cdot 5^2 \cdot 7^2$

The formula for the number of factors is: Number of factors $= (e_1 + 1)(e_2 + 1)(e_3 + 1)\cdots$ where $e_1, e_2, e_3, \ldots$ are the exponents of the prime factors.

Here, the exponents are:
• $e_1 = 4$ (for $2^4$),
• $e_2 = 3$ (for $3^3$),
• $e_3 = 2$ (for $5^2$),
• $e_4 = 2$ (for $7^2$).

Thus, the number of factors is: $(4+1)(3+1)(2+1)(2+1) = 5 \cdot 4 \cdot 3 \cdot 3$

Now, calculate: $5 \cdot 4 = 20$, $20 \cdot 3 = 60$, $60 \cdot 3 = 180$

**Final Answer:** $\boxed{180}$ |

**Table F.8 – continued from previous page**

| Dataset | Prompt | Ground Truth | Pred (Ours) | Pred (GPT-4o) |
|---|---|---|---|---|
| **Counting-Probability** | Solve the following competition math problem: What is the value of $\frac{9!\cdot5!\cdot2!}{8!\cdot6!}$? | We do some simplification, taking advantage of the fact that $n! = n \cdot (n-1)!$: $$\frac{9! \cdot 5! \cdot 2!}{8! \cdot 6!}$$ $$= \frac{9 \cdot 8! \cdot 5! \cdot 2}{8! \cdot 6 \cdot 5!}$$ $$= \frac{9 \cdot 2}{6} = \boxed{3}.$$ | To find the height of the cylindrical water tank, we can use the formula for the volume of a cylinder: We are given: $$\frac{9!\cdot5!\cdot2!}{8!\cdot6!}$$ **Step 1: Simplify the factorials** Numerator: $9! \cdot 5! \cdot 2!$ Denominator: $8! \cdot 6!$ **Step 2: Cancel common factorials** Notice that $9! = 9 \cdot 8!$, so we can cancel $8!$: $\frac{9\cdot8!\cdot5!\cdot2!}{8!\cdot6!} = \frac{9\cdot5!\cdot2!}{6!}$ Now, recall that $6! = 6 \cdot 5!$, so: $\frac{9\cdot5!\cdot2!}{6\cdot5!} = \frac{9\cdot2!}{6}$ Since $2! = 2$, we have: $\frac{9\cdot2}{6} = \frac{18}{6} = 3$ **Final Answer:** $\boxed{3}$ ✓**Verification:** Numerator: $362880 \cdot 120 \cdot 2 = 87091200$ Denominator: $40320 \cdot 720 = 28819200$ Now compute: $87091200/28819200 = 3$ Matches. ✓ | We are tasked with evaluating the expression: $\frac{9!\cdot5!\cdot2!}{8!\cdot6!}$ **Step 1: Simplify the factorials** Recall that $n! = n \cdot (n-1)!$. Simplify $9!$ and $8!$: $\frac{9!}{8!} = 9$. **Step 2: Rewrite the expression** Substituting the simplifications, the expression becomes: $\frac{9\cdot5!\cdot2!}{6!}$ **Step 3: Expand $5!$ and $2!$** $5! = 120, \quad 2! = 2$. Simplify $6! = 720$. Now the expression becomes: $\frac{9\cdot120\cdot2}{720}$ **Step 4: Simplify the numerator and denominator** First, calculate the numerator: $9 \cdot 120 \cdot 2 = 2160$. Now divide by the denominator: $\frac{2160}{720} = 3$ **Final Answer:** $\boxed{3}$ |

**Table F.8 – continued from previous page**

| Dataset | Prompt | Ground Truth | Pred (Ours) | Pred (GPT-4o) |
|---|---|---|---|---|
| **Geometry** | Solve the following competition math problem: A recipe for crispy rice treats results in a mixture that fills a 9-inch by 13-inch pan to a depth of one inch. If a crispy rice treats mixture resulting from 1.5 times the original recipe is poured into a pan that is 10 inches by 15 inches, to what depth, in inches, will the pan be filled? Express your answer as a decimal to the nearest hundredth. | The volume of crispy rice treats resulting from the original recipe is $9 \cdot 13 \cdot 1 = 117$ cubic inches. Thus the volume obtained by making 1.5 times the original recipe is $1.5 \cdot 117 = 175.5$ cubic inches. So the depth to which the pan is filled is: $\frac{175.5}{10 \cdot 15} = \boxed{1.17}$ inches. | We are given the following:
• Original recipe fills a 9-inch by 13-inch pan to a depth of 1 inch.
• We now have **1.5 times the original recipe**.
• This is poured into a **10-inch by 15-inch pan**.
• We are to find the **depth** to which this mixture fills the new pan.

**Step 1: Compute volume of original mixture**
The original mixture fills a pan that is:
• 9 inches wide
• 13 inches long
• 1 inch deep
So volume is:
$V = 9 \times 13 \times 1 = 117$ cubic inches
Now, we have **1.5 times this amount**:
$1.5 \times 117 = 175.5$ cubic inches
This volume is poured into a pan that is **10 inches by 15 inches**.
Let $h$ be the depth (in inches) to which this 175.5 cubic inches fills the pan. So:
$10 \times 15 \times h = 175.5$
Solve for $h$:
$150h = 175.5$
$h = \frac{175.5}{150} = 1.17$

**Final Answer:**
$\boxed{1.17}$
The depth to which the pan is filled is **1.17 inches**. ✓

**Verification:**
• Original volume: $9 \times 13 = 117$
• 1.5 times: $1.5 \times 117 = 175.5$
• New pan: $10 \times 15 = 150$, so $h = 175.5/150 = 1.17$
Correct. ✓
$\boxed{1.17}$
Perfect! ✓

**Final Answer:**
$\boxed{1.17}$ inches | We are tasked with finding the depth of the crispy rice treats mixture in a 10-inch by 15-inch pan when 1.5 times the original recipe is used.

**Step 1: Calculate the volume of the original recipe**
The original recipe fills a 9-inch by 13-inch pan to a depth of 1 inch. The volume of the mixture is:
Volume $=$ length $\times$ width $\times$ depth
$= 9 \times 13 \times 1 = 117$ cubic inches.

**Step 2: Calculate the volume of 1.5 times the original recipe**
If 1.5 times the original recipe is used, the volume becomes:
$1.5 \times 117 = 175.5$ cubic inches.
**Step 3: Calculate the depth in the new pan**
The new pan has dimensions 10 inches by 15 inches. Let $d$ represent the depth of the mixture in this pan. The volume of the mixture in the new pan is: Volume $=$ length $\times$ width $\times$ depth. Substituting the known values:
$175.5 = 10 \times 15 \times d$.
Simplify:
$175.5 = 150d$.
Solve for $d$:
$d = \frac{175.5}{150} = 1.17$ inches.

**Final Answer:**
**1.17 inches**. |

**Table F.8 – continued from previous page**

| Dataset | Prompt | Ground Truth | Pred (Ours) | Pred (GPT-4o) |
|---------|--------|--------------|-------------|---------------|
| **Geometry** | Solve the following competition math problem: Point $A$ has coordinates $(x, 6)$. When Point $A$ is reflected over the $y$-axis it lands on Point $B$. What is the sum of the four coordinate values of points $A$ and $B$? | The coordinates of point $B$ are $(-x, 6)$. The sum of all four coordinates is $x+6+(-x)+6 = \boxed{12}$. | To solve the problem, we need to understand the effect of reflecting a point over the $y$-axis.

**Step 1: Reflect Point $A$ over the $y$-axis**

Point $A$ has coordinates $(x, 6)$. When a point is reflected over the $y$-axis, the $x$-coordinate changes sign, while the $y$-coordinate remains the same.
So, the reflection of point $A(x, 6)$ over the $y$-axis is:
$B = (-x, 6)$

**Step 2: List the coordinates of both points**
• Point $A$: $(x, 6)$
• Point $B$: $(-x, 6)$

**Step 3: Sum all four coordinate values**
We sum the $x$-coordinates and $y$-coordinates separately:
$x + (-x) + 6 + 6 = 0 + 12 = 12$

**Final Answer:**
$\boxed{12}$
This result is independent of the value of $x$. The sum of the four coordinate values is always **12**, regardless of the original $x$-coordinate. ✓

✓**Final Answer:**
$\boxed{12}$
This is the final answer, obtained by careful reflection and coordinate addition. No other interpretation or simplification is needed. ✓✓✓

**Alternate Check:**
Let's verify with a specific example.
Let $x = 5$. Then:
• $A = (5, 6)$
• $B = (-5, 6)$
Sum of coordinates:
$5 + (-5) + 6 + 6 = 0 + 12 = 12$
Same result! This confirms our general solution is correct.
$\boxed{12}$ ✓✓✓ | To solve the problem, we start by understanding the reflection of a point over the $y$-axis. When a point $(x, y)$ is reflected over the $y$-axis, its new coordinates become $(-x, y)$.

**Step 1: Coordinates of Point $A$**
Point $A$ has coordinates $(x, 6)$.

**Step 2: Coordinates of Point $B$**
When Point $A$ is reflected over the $y$-axis, its new coordinates become $(-x, 6)$. Thus, Point $B$ has coordinates $(-x, 6)$.

**Step 3: Sum of the Four Coordinate Values**
The four coordinate values are:
• From Point $A$: $x$ and 6,
• From Point $B$: $-x$ and 6.
The sum of these values is:
$x + 6 + (-x) + 6 = 6 + 6 = 12.$

**Final Answer:**
$\boxed{12}$ |

**Table F.8 – continued from previous page**

| Dataset | Prompt | Ground Truth | Pred (Ours) | Pred (GPT-4o) |
|---|---|---|---|---|
| **CW** | Tropeday # 2.1 - Noblesse Oblige | The guards at the main entrance had a formal demeanor... The Decian bowed. "Lady Kumi, Master d'Ours. A pleasure." The senior officer's face darkened. "Lord Asano will speak with him in an hour... Wear your best boy." Kumi Asano made a face. "Decian Areto, your manners. I would have you treat him as you would me, with honor and respect." d'Ours turned to his friend with rueful amusement. "Perhaps... but most take one look at me... I can never be completely a part of this tribe." 

 *(...d'Ours enters the court...)* Lord Asano Naoya sat cross legged... d'Ours knelt. "Hail Lord Asano... How may I serve." "Are you well, Albatross d'Ours? And where do your loyalties lie?" "With you of course, my lord. And the clan as a whole." "And what is duty compared to death?" "Duty is heavier than a mountain, my lord. Death is lighter than a feather." The noble lord paused. "I am certain in my choice... I would be honored to have you serve me." | The palace smelled of lavender and old wood... Elara stepped through the grand hall. She wore the blue sash of the Matriarch... Tradition demanded it. Duty required it. They called her the Moonlit Queen... When the northern provinces burned, she did not send soldiers. She sent herself. She stood before the Council Chamber. "We cannot sustain this," said Lord Varrin. "The people are starving." "And yet," said Elara, "we spend more on feasts than famine relief. I propose that we divert half the royal budget... I will sell the northern hunting lodge... Noblesse oblige, my lords. It is not just a phrase. It is a debt we owe to those who come after us." 

 That night, Elara walked to the edge of the palace garden. She removed the blue sash and laid it on a stone bench. She would not wear it again. Because noblesse oblige was not a title. It was a choice. And some choices required more than a crown. | You stop in your tracks, heart thudding as your eyes meet theirs. They look surprised but not unkind, and for a moment, silence stretches between you like a fragile thread. "Hey," you begin awkwardly... "I know it's been years, but I... I just want to say I'm sorry. For everything." Their expression softens... "Yeah, I've thought about it too. I guess we were both figuring things out." You both laugh... and you feel the years begin to melt away. Finally, you add, "And for the record, I really didn't mean to spill red wine on your mom's wedding dress." |

**Table F.8 – continued from previous page**

| Dataset | Prompt | Ground Truth | Pred (Ours) | Pred (GPT-4o) |
|---|---|---|---|---|
| **EU** | On top of his old powers , Spider Man suddenly acquires the ability to control actual spiders . The crime rate in New York has reduced drastically since then . | **GOON 1:** "I mean, ick, ya' know?" **GOON 2:** "Yeah, man." **GOON 1:** "It was bad enough when he was beating our mugs and then sticking us with that gooey crap, but this is some next-level creep factor." **GOON 2:** "I know what you mean. Jimmy and me tried to pull one over on a gas station last month... his creepy-ass little friends were not appreciated." **GOON 1:** "Oh yeah, Jimmy says he found a few of them in his shoe a week later. Like.. that isn't cool, man." 

 **GOON 2:** "I can stand getting beaten up... but this... who even thinks up this kind of stuff?" **GOON 1:** "I ain't gonna say it to his face though..." **GOON 1:** "...... Mahoney left last night. Some guys said they saw like a million spiders crawl into his car..." **GOON 2:** "Damn. You know they say the Spider never goes to Jersey." **GOON 1:** "I'm with ya'. But let's stop by the quik-mart first for some soap..." | The sun dipped below the skyline of Midtown... Peter Parker adjusted his spider-shaped sunglasses. It wasn't the webbing or speed that drew the most attention now—it was the quiet, deliberate way he moved among the brick walls... He wasn't just climbing anymore. He was *calling* them. 

 The first time he did it was during a mugging on West 4th... Peter didn't pull out a web shooter. He reached for the small device at his belt and whispered, "Come." From the ivy, a dozen brown garden spiders emerged... forming a thin line across the alleyway. The thief tripped... and a second wave swarmed over him. "Whoa, whoa!" he stammered. "Stealing?" Peter stepped forward. "You think power means force. But strength is control." 

 The thief froze. Around him, the spiders began to rearrange themselves... A single leader climbed onto Peter's shoulder... It was *talking*. And Peter listened. That night, crime statistics across New York dropped by 15%... Word spread fast. At first, people joked... But then came the silence. The eerie, itchy silence that settled over the alleyways. Criminals stopped watching the skies... Peter Parker sat on a rooftop edge, watching a thousand tiny eyes watch the city back. He didn't need to be everywhere anymore. The neighborhood was finally watching itself. ✓ | Spider-Man's new-found ability to control actual spiders would undoubtedly amplify his crime-fighting capabilities. The psychological impact alone—thousands of spiders swarming criminals—would be a powerful deterrent. This could explain the drastic reduction in New York's crime rate, as criminals likely fear the prospect of facing not just Spider-Man, but an army of arachnids. It's a creative evolution of his powers, cementing his role as a more intimidating and effective protector of the city. |

| Dataset | Prompt |
|---|---|
|  | Question: What kind of coating has been used? Explain the reasoning behind the answer. |

| Ground Truth | Pred. (Ours) | Pred. (GPT4o) |
|---|---|---|
| The coating is made of sugar and glistening in the light. | The image shows a box filled with various types of donuts, including glazed, chocolate, and sprinkled ones. The glazed donuts have a shiny, glossy appearance, while the chocolate donuts show a dark, chocolatey coating. The sprinkled donut has colorful, small pieces on top. The glazed donuts are typically coated with a sugar glaze, which gives them a glossy sheen. The chocolate donuts are coated with chocolate, which appears dark and smooth. The sprinkled donut is topped with colorful sprinkles.

The glazed donuts have a glossy, shiny coating, likely from a sugar glaze, while the chocolate donuts show a dark, smooth chocolate coating. The sprinkled donut has colorful, small pieces on top. The glazed donuts are typically coated with a sugar glaze, and the chocolate donuts are coated with chocolate.

The glazed donuts are coated with a sugar glaze, which gives them a glossy sheen. | The donuts in the image have various coatings:

1. Top left: Glazed coating.
2. Top middle: Glazed coating.
3. Top right: Sprinkles on a white frosting base.
4. Bottom left and middle: Chocolate frosting.
5. Bottom right: Caramel or butterscotch frosting.

The coatings are identifiable by their color, texture, and appearance. |

*Figure F.5.* Visualization of experimental results (Part 1).

| Dataset | Prompt |
|---|---|
|  | Question: Where is the sun with respect to the person wearing red coat? Explain the reasoning behind the answer. |

| Ground Truth | Pred. (Ours) | Pred. (GPT4o) |
|---|---|---|
| The sun is behind the man with the coat given his shadow. | The person in the red coat is standing on a snowy slope, and the sun is positioned behind them, casting a shadow on the snow. The shadow is visible on the snow, and the sun is likely low on the horizon, as the sky is clear and the mountains are visible in the background. The sun is not directly overhead, which would cast no shadow. The shadow is cast toward the viewer, indicating the sun is to the left or behind the person.

The sun is positioned behind the person, casting a shadow on the snow. The shadow is visible on the snow, and the sun is likely low on the horizon, as the sky is clear and the mountains are visible in the background. The sun is not directly overhead.

The sun is behind the person, casting a shadow on the snow. | The sun is in front of the person wearing the red coat, slightly to their left. This is evident from the shadow cast by the person, which extends behind them and slightly to their right. |

*Figure F.6.* Visualization of experimental results (Part 2).

| Dataset | Prompt |
|---|---|
|  | Question: The tourists are hoping to get pictures of the giraffe in its natural? Explain the reasoning behind the answer. |

| Ground Truth | Pred. (Ours) | Pred. (GPT4o) |
|---|---|---|
| A single white ban is driving down a safari. a giraffe is roaming around in the field. | The image shows a giraffe walking across a grassy plain, with a van and a car nearby, suggesting a safari or wildlife tour. The giraffe is positioned in the foreground, and the tourists are seated in the vehicles, likely observing the animal. The presence of a van with a roof rack and a car with passengers implies a guided tour, and the giraffe is the primary focus. The tourists are likely hoping to capture the moment, especially since the giraffe is relatively close and the lighting is favorable.

The tourists are likely hoping to get pictures of the giraffe in its natural environment, as the giraffe is the main subject and the vehicles are positioned for observation. The giraffe is walking across a grassy plain, and the tourists are seated in the vehicles, suggesting a guided tour. The setting appears to be a safari or wildlife park. | Yes, the tourists are hoping to get pictures of the giraffe in its natural habitat.

The reasoning is based on the image showing a safari vehicle with tourists observing and photographing a giraffe in an open, natural savanna-like environment, which is typical of the giraffe's habitat. |

*Figure F.7.* Visualization of experimental results (Part 3).

| Dataset | Prompt |
|---------|--------|
|  | Question: What type feet do these birds have? Explain the reasoning behind the answer. |

| Ground Truth | Pred. (Ours) | Pred. (GPT4o) |
|--------------|--------------|---------------|
| These birds are ducks and ducks are known to have webbed feet. | The image shows a large group of waterfowl, including ducks and geese, floating on calm water. Many of them are submerged or partially submerged, with their legs extended or partially underwater. The birds in the foreground, such as the one with a yellow beak, are fully in the water, while others are partially submerged. This suggests they have webbed feet, which are common in waterfowl for efficient swimming and wading. The presence of both submerged and partially submerged birds supports the idea that they are adapted to water environments.

The birds in the image, including ducks and geese, have webbed feet, which allow them to swim and wade in water. The image shows them floating and submerged, with legs extended or partially underwater, consistent with webbed foot anatomy. The presence of both submerged and partially submerged birds supports the idea that they are adapted to water environments.

Final answer: These birds have webbed feet. | These birds have webbed feet.

Reasoning: The birds in the image are waterfowl, which are adapted for swimming. Webbed feet are a common feature in waterfowl, as they help propel them efficiently through the water. |

*Figure F.8.* Visualization of experimental results (Part 4).

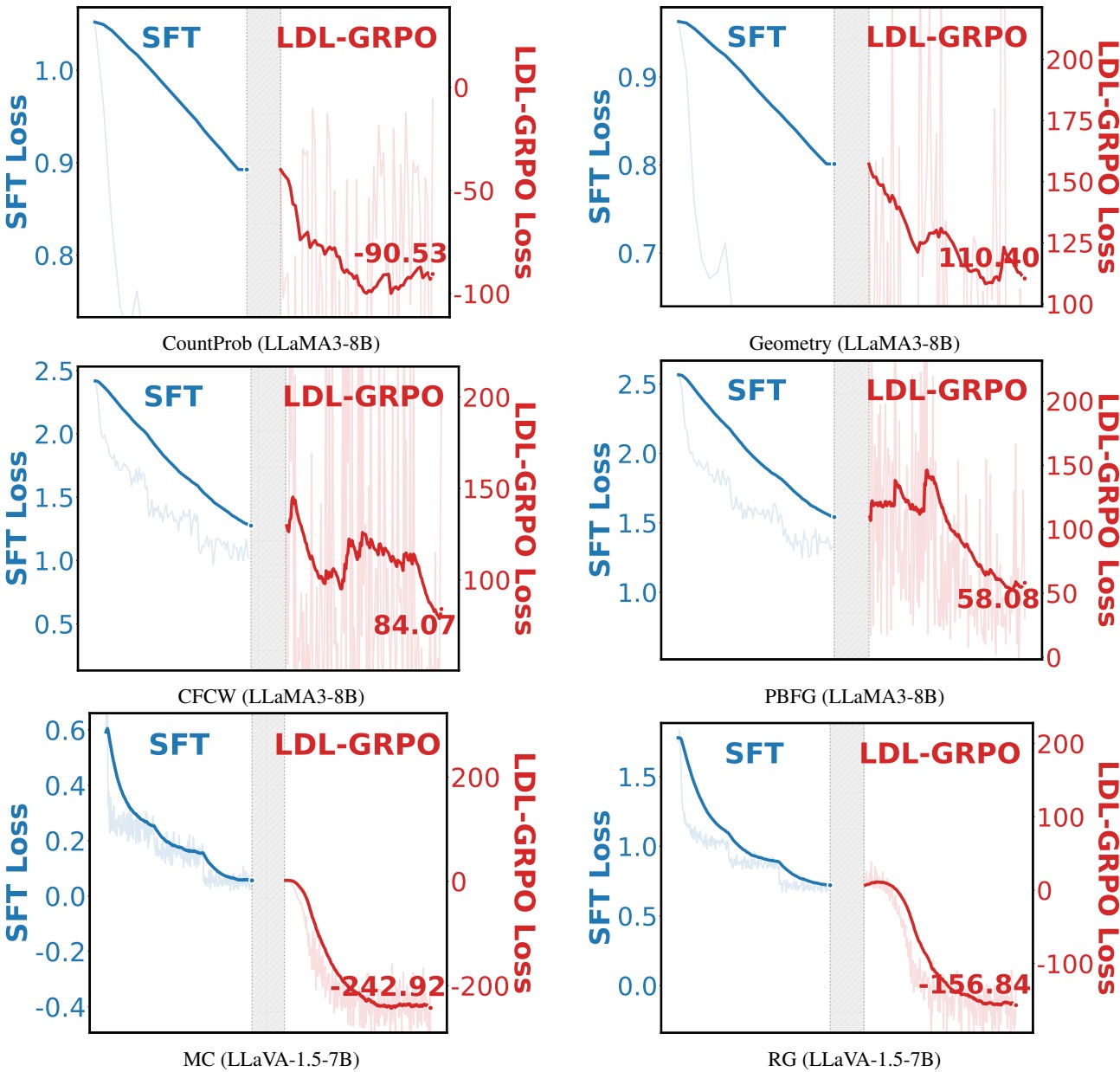

*Figure F.9.* **Two-stage convergence from SFT to LDL-GRPO across models and tasks.** Each panel shows one model–task pair, illustrating a stable transition from SFT to LDL-GRPO.

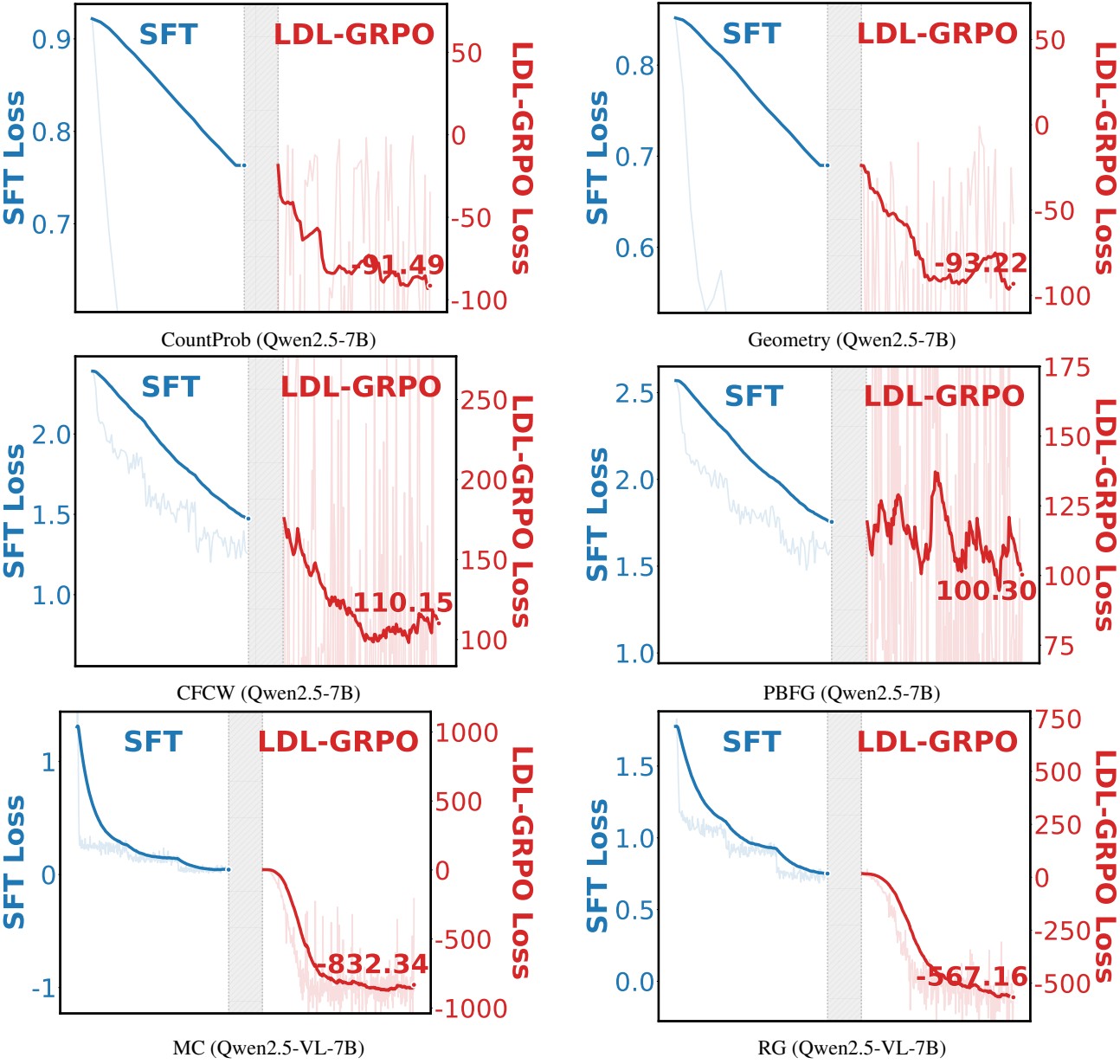

*Figure F.10.* **Two-stage convergence from SFT to LDL-GRPO across models and tasks.** Each panel shows one model–task pair, illustrating a stable transition from SFT to LDL-GRPO.

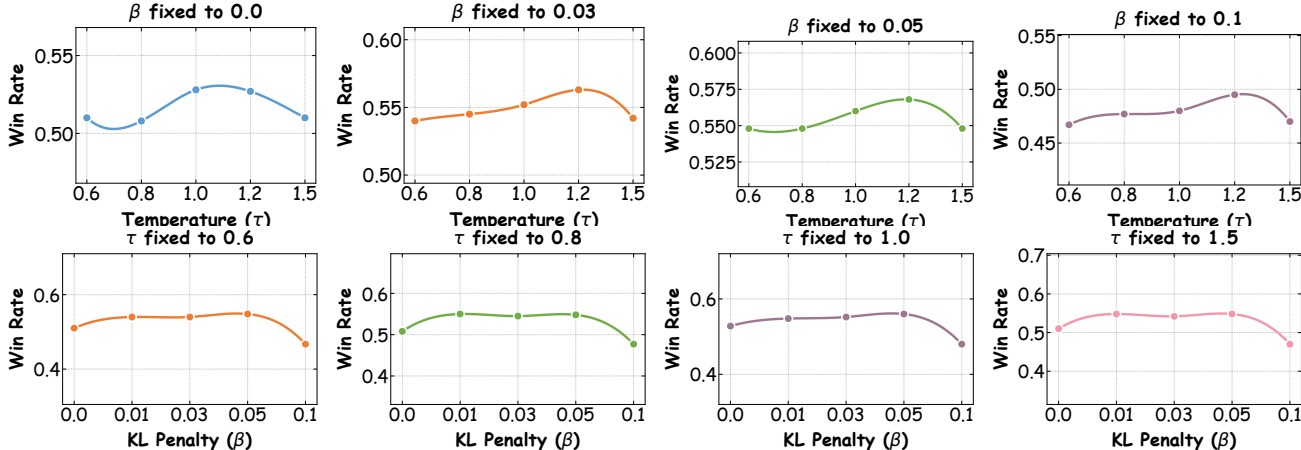

*Figure F.11.* **Compact sensitivity plots on CountProb (LLaMA3-8B).** Top: fixed-$\beta$ sweeps; bottom: fixed-$\tau$ sweeps.

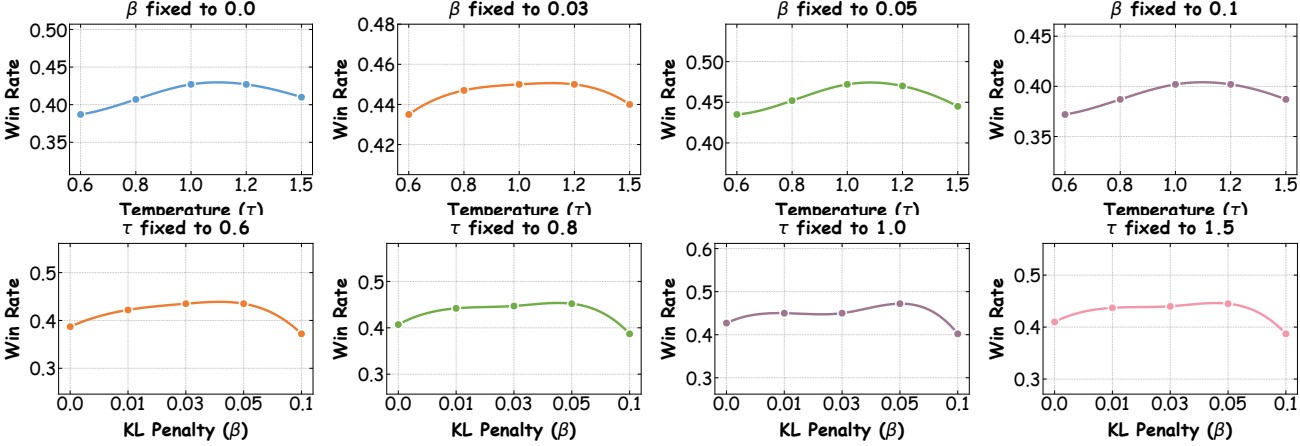

*Figure F.12.* **Compact sensitivity plots on A-OKVQA-RG (LLaVA-7B).** Top: fixed-$\beta$ sweeps; bottom: fixed-$\tau$ sweeps.

