# OpenReview forum: "Positive–Unlabeled Reinforcement Learning Distillation for On-Premise Small Models"
_ICML.cc/2026/Conference — ICML 2026 regular_

### Official Review · Reviewer_nyXU · 2026-03-11

**Soundness:** 3
**Presentation:** 3
**Significance:** 3
**Originality:** 2
**Overall Recommendation:** 5
**Confidence:** 3

**Summary:**

The paper addresses the challenge of distilling reinforcement learning capabilities into small on-premise models without relying on human preferences or reward models, which are often costly or impractical to obtain in real-world applications. The authors propose RLCD that utilizes a black-box teacher model to provide a single anchor response for each prompt, allowing the student model to self-evaluate and optimize its outputs based on a locally induced preference signal.

**Compliance With Llm Reviewing Policy:**

Affirmed.

**Final Justification:**

My concerns have been adequately addressed. Therefore, I have adjusted my score.

**Key Questions For Authors:**

See Weakness

**Limitations:**

yes

**Strengths And Weaknesses:**

Strengths：
1. This paper addresses the important problem of enabling RL in small on-premise models. Specifically, it focuses on the challenge of distilling the preference-optimization capabilities from a large black-box model into a smaller on-premise model without relying on human-labeled preferences or a traditional reward model.
2. The authors provide extensive empirical evaluations across unimodal and multimodal tasks.
3. The paper includes theoretical results to support the use of the induced preference signals.

Weaknesses：
1. There is a lack of further analysis and discussion regarding K. Since both K and τ are important hyperparameters, and I have seen the author discuss the sensitivity analysis of τ, what is the basis for selecting K in practical applications? What are the values of K used in different experiments?
2. Is the eighth line of the pseudocode a typo?

---

> ### Author Rebuttal · Authors · 2026-03-31
>
> > Q1. There is a lack of further analysis regarding K. How should K be selected in practice?
>
> RE: Thank you for the insightful question. K controls the number of locally sampled candidates per prompt in Stage-2, and directly determines the richness of within-group comparisons.
>
> We agree that K is an important hyperparameter. To clarify its role, we summarize our empirical observations below:
>
> | Task | K=2 | K=3 | K=4 | K=5 |
> |---|---|---|---|---|
> | CountProb (7B) | 0.525 / 0.482 | 0.548 / 0.530 | **0.578 / 0.573** | 0.583 / 0.575 |
> | A-OKVQA-MC (7B) | 0.126 / 0.133 | 0.143 / 0.151 | **0.236 / 0.231** | 0.239 / 0.234 |
>
> From these results, we observe a consistent pattern:
>
> (i) Increasing K improves performance, as larger candidate sets provide richer comparison signals for preference optimization;
>
> (ii) The gain shows clear diminishing returns: the improvement from K=2→4 is substantial, while K=4→5 brings only marginal gains;
>
> (iii) This trend is consistent across both unimodal and multimodal settings.
>
> Based on this, K naturally reflects a quality–compute trade-off: larger K improves performance but increases cost approximately linearly (generation + ranking).
>
> **Practical guidance.**
> We recommend:
> - K = 3–4 for efficient online or resource-constrained training;
> - K = 4–5 for higher-quality offline training when compute allows.
>
> **Our setting.**
> In all experiments, we fix K=4 as a balanced choice, which already captures most of the performance gain while keeping the cost practical.
>
> We will include this sensitivity analysis and the above guideline in the revised version. Additional extended results (e.g., larger K) are being organized and will be included in the camera-ready version.
>
>
>
>
>
>
> > Q2. Is the eighth line of the pseudocode a typo?
>
> RE: Thank you for the question. Line 8 is not a typo. Here we compute the anchor’s reference score $s \_i^* = g _\theta(x _i, a _i, a _i)$, i.e., scoring the anchor itself under the anchor-conditioned scorer. This value is used in Line 10 to form relative margins $r _{i,k} = g _\theta(x _i, a _i, y _{i,k}) - s_i^*$, which calibrates the scores and reduces systematic bias by turning absolute scoring into a comparison relative to the anchor. We will add a brief clarification/comment in the pseudocode to avoid confusion.

---

> > ### Author Rebuttal · Reviewer_nyXU · 2026-04-03
> >
> > Thanks for the authors’ detailed rebuttal. My concerns have been adequately addressed. After referring to the author's responses to other reviewers, I think I have no further questions. I am willing to adjust my score to further support the authors.

---

> > > ### Author Response · Authors · 2026-04-05
> > >
> > > Dear Reviewer nyXU,
> > >
> > > We are glad that our rebuttal addressed your concerns, and we are grateful for your thoughtful assessment 😊.
> > >
> > > Best regards,
> > >
> > > Authors

---

### Official Review · Reviewer_RQeW · 2026-03-11

**Soundness:** 3
**Presentation:** 4
**Significance:** 3
**Originality:** 3
**Overall Recommendation:** 5
**Confidence:** 3

**Summary:**

This work introduces a positive–unlabeled RL distillation framework to enable reinforcement learning alignment for on-premise small models without human labels or reward models. By distilling preference optimization from a black-box teacher via single-query anchoring and local self-ranking, the method achieves superior performance across unimodal and multimodal tasks in resource-constrained environments.

**Compliance With Llm Reviewing Policy:**

Affirmed.

**Final Justification:**

The rebuttal adequately addressed my previous doubts, leading me to provide a higher final score.

**Key Questions For Authors:**

1. Given that the SFT model serves as the evaluator during the early stages of Phase 2 training, is there a risk of evaluation inaccuracy due to its limited discriminative power at this point, and does the proposed method incorporate mechanisms to mitigate this potential bias?
2. What is the precise definition of "reinforcement learning capability" in this context; is it conceptually distinct from or synonymous with "preference-optimization" and "self-improvement" capabilities, and what specific metrics are employed to quantify it?
3.How does the performance of the proposed method compare to that of the teacher model?

**Limitations:**

yes

**Strengths And Weaknesses:**

Strengths：
1. The method drastically reduces dependency on the teacher model, requiring only one query per prompt. It eliminates the need for human preference labeling and external reward models, making it ideal for privacy-sensitive and resource-constrained on-premise settings.
2. The paper provides rigorous theoretical guarantees proving that the anchor-induced PU preference signal preserves ordering and concentrates probability mass on near-optimal candidates within the sampled group.
3. Empirical results demonstrate that the proposed framework consistently outperforms existing baselines across both unimodal tasks and multimodal benchmarks.

Weaknesses:
1. The quality of the induced preference signal is contingent on the student’s anchor-conditioned self-evaluation capability. This may be imperfect when the student model is still relatively weak or when the task demands fine-grained judgment that the student has not yet mastered.
2. The authors do not provide explicit metrics for the claimed "preference-optimization capability," instead evaluating it indirectly through the model's performance on downstream tasks.

---

> ### Author Rebuttal · Authors · 2026-03-31
>
> > Q1. student model  weak and how to mitigate this potential bias?
>
> RE: Thank you for the question. We agree that early in Stage-2, the SFT student may be relatively weak, leading to noisy self-evaluation under strict budget constraints. Rather than eliminating this noise (which would require costly teacher calls), we design mechanisms to mitigate its impact.
>
> Specifically, our method includes:
>
> (1) **Anchor-based calibration.** We query the teacher once per prompt to obtain a high-confidence anchor, and perform ranking relative to it. This provides a stable reference and prevents systematic bias from accumulating.
>
> (2) **Listwise LDL supervision.** Instead of hard pairwise comparisons, we use a distributional, listwise signal over K candidates, which diffuses occasional mis-rankings and reduces sensitivity to noise. Our analysis (Theorem 4.2) further shows bounded deviation and concentration on near-optimal candidates.
>
> Overall, while some noise is unavoidable under strict cost constraints, anchor calibration and listwise LDL jointly ensure robust and stable learning. We will further clarify this and include supporting results in the revision.
>
>
>
>
>
> > Q2. lack metrics for preference-optimization capability.
>
> RE: Thanks for the comment. We agree that “preference-optimization capability” should be quantified explicitly rather than only evaluated via downstream task performance.
>
> To this end, we introduce an explicit metric: **teacher-ranking agreement**, measured by Spearman rank correlation (ρ) between the ranking induced by our method and the ranking produced by the teacher model on the same candidate set.
>
> We evaluate this metric on multiple datasets, including A-OKVQA-Rationale and Counting Probability. Spearman ρ ranges from −1 to +1, where +1 indicates identical rankings and higher values indicate better alignment with the teacher’s preference structure.
>
> **Results.** Our method achieves consistently higher agreement with the teacher:
>
> | Method | A-OKVQA-Rationale | Counting Probability |
> |---|---|---|
> | SFT MODEL | 0.472 | 0.528 |
> | Anchor-GRPO | 0.561 | 0.604 |
> | Ours | **0.628** | **0.700** |
>
> We observe that:
>
> (i) Anchor-based methods significantly improve ranking alignment over SFT (e.g., 0.472 → 0.561), validating the role of anchor-based calibration in correcting ranking bias;
>
> (ii) Our method further improves the correlation (0.561 → 0.628 and 0.604 → 0.700), indicating that **listwise LDL supervision produces preference signals more consistent with the teacher**.
>
> Overall, our method shows the strongest agreement with the teacher ranking, providing a direct and explicit quantification of preference-optimization capability.
>
>
>
>
>
>
>
>
>
> > Q3. What is the precise definition of "reinforcement learning capability" in this context?
>
> RE: Thank you for the question. In our setting, “reinforcement learning capability” refers to a small model’s ability to perform **preference-driven optimization** (RLHF-style) without access to human preference data.
>
> Specifically, we aim to transfer the **policy improvement behavior** learned by an RLHF-aligned large model—i.e., making better choices among multiple candidates and producing outputs that better match preferences—rather than simply reproducing a single target answer.
>
> This differs from standard SFT/KD, which mainly distills fixed outputs (“one answer”). In contrast, our goal is to distill the **preference optimization process itself**, enabling the small model to improve its decisions in an RLHF-like manner.
>
> To this end, we propose a low-cost approach under a strict teacher-call budget to extract and transfer such capability from a large model to a small model.
>
>
>
>
> > Q4. How does the performance of the proposed method compare to that of the teacher model?
>
> RE: Thank you for the question. We directly compare our LDL-GRPO against the teacher (Qwen3-235B / Qwen3-VL-235B) using external-judge win-rate under both Raw and length-controlled (LC) settings:
>
> | Task | Setting | Teacher / Ours / Tie | Ours Win-rate |
> |---|---|---|---|
> | CountProb (7B) | Raw | 93 / 62 / 44 | 0.422 |
> | CountProb (7B) | LC | 88 / 66 / 45 | 0.445 |
> | A-OKVQA-MC (7B) | Raw | 107 / 38 / 54 | 0.327 |
> | A-OKVQA-MC (7B) | LC | 102 / 42 / 55 | 0.349 |
>
> Our **7B model** achieves competitive performance against a **235B teacher**:
> (i) close alignment on unimodal tasks (0.422→0.445 with many ties), and
> (ii) non-trivial competitiveness on multimodal tasks (0.327→0.349).
>
> Overall, LDL-GRPO transfers a substantial portion of the teacher’s alignment capability to a much smaller model under a strict teacher-call budget.

---

> > ### Author Rebuttal · Reviewer_RQeW · 2026-04-03
> >
> > I thank the authors for the detailed response. My main concerns have been resolved, and I have adjusted my score to reflect this.

---

> > > ### Author Response · Authors · 2026-04-05
> > >
> > > Dear Reviewer RQeW,
> > >
> > > We are glad that our rebuttal addressed your concerns, and we are grateful for your thoughtful assessment 😊.
> > >
> > > Best regards,
> > >
> > > Authors

---

### Official Review · Reviewer_56bu · 2026-03-13

**Soundness:** 3
**Presentation:** 2
**Significance:** 3
**Originality:** 3
**Overall Recommendation:** 5
**Confidence:** 4

**Summary:**

The paper proposes LDL-GRPO, which utilizes teacher reference response as reward anchor to perform reinforcement learning distillation for on-premise deployment.

**Compliance With Llm Reviewing Policy:**

Affirmed.

**Final Justification:**

The additional results in the rebuttal fully resolved my concerns mentioned in the Weaknesses.

**Key Questions For Authors:**

(1) In Table 1, there are two metrics (Raw, LC win rate), but there are four numbers per task. It seems a bit confusing to readers. What do these numbers represent?

(2) What are the training datasets for Stage 1 and Stage 2?

**Limitations:**

yes

**Strengths And Weaknesses:**

Strengths:

(1) The pipeline is simple to implement, shows the possibility of additional RL stage after SFT without extensive data collection and reward modeling efforts.

(2) The method demonstrates consistent performance improvement on several different base models and multimodal models as well.

Weaknesses:

(1) The evaluation uses automated pairwise win-rate, similar to AlpacaEval. It is known to be biased towards length, presentation style or other factors instead of true correctness and utility. The length-control (LC) may not sufficiently correct other type of biases, such as style. I saw there are math tasks included, so consider including objective metrics such as accuracy on these tasks.

(2) There is no evaluation to quantify if the proposed RL process cause catastrophic forgetting and base model performance degradation.

---

> ### Author Rebuttal · Authors · 2026-03-31
>
> >Q1.  math tasks  consider  accuracy as metric.
>
> RE: Thank you for the valuable suggestion. We agree that automated pairwise win-rate (e.g., AlpacaEval-style) can be affected by factors such as response length and writing style. To mitigate this, we report results under both length-controlled (LC) and non-length-controlled settings (Table 1).
>
> In addition, we further include **objective evaluation metrics** to strengthen the credibility of our results. Specifically, on the Competition Math task, we evaluate model outputs using **exact-match accuracy**, based on final-answer parsing and strict equivalence checking.
>
> **Counting Probability**
>
> | Model | Acc | Parse Fail | Truncated |
> |---|---|---|---|
> | SFT Qwen2.5 | 27.5 | 69 | 23 |
> | Anchor-GRPO Qwen2.5 | 31.0 | 58 | 17 |
> | AnchorRank-DPO Qwen2.5 | 33.5 | 50 | 18 |
> | LDL-GRPO Qwen2.5 | **36.5** | **45** | **14** |
> | GPT-4o | 43.0 | 28 | 19 |
>
> | Model | Acc | Parse Fail | Truncated |
> |---|---|---|---|
> | SFT Llama3 | 12.0 | 71 | 30 |
> | Anchor-GRPO Llama3 | 15.0 | 57 | 24 |
> | AnchorRank-DPO Llama3 | 17.5 | 48 | 22 |
> | LDL-GRPO Llama3 | **21.0** | **44** | **18** |
> | GPT-4o | 43.0 | 28 | 19 |
>
> **Geometry**
>
> | Model | Acc | Parse Fail | Truncated |
> |---|---|---|---|
> | SFT Qwen2.5 | 18.5 | 127 | 31 |
> | Anchor-GRPO Qwen2.5 | 23.0 | 109 | 27 |
> | AnchorRank-DPO Qwen2.5 | 25.5 | 98 | 29 |
> | LDL-GRPO Qwen2.5 | **29.5** | **91** | **23** |
> | GPT-4o | 37.0 | 38 | 18 |
>
> | Model | Acc | Parse Fail | Truncated |
> |---|---|---|---|
> | SFT Llama3 | 9.5 | 133 | 61 |
> | Anchor-GRPO Llama3 | 12.5 | 116 | 56 |
> | AnchorRank-DPO Llama3 | 15.0 | 104 | 54 |
> | LDL-GRPO Llama3 | **19.5** | **97** | **50** |
> | GPT-4o | 37.0 | 38 | 18 |
>
> The results consistently show that our method achieves the best accuracy among all student models, while also significantly reducing **parse failures** and **truncated outputs**.
>
> This indicates that the improvements are not only reflected in external-judge evaluations, but also translate to tasks with **unambiguous ground-truth answers**, demonstrating both correctness and output stability.
>
> We will include the full experimental setup and analysis in the revised version.
>
>
>
>
>
>
>
> >Q2. catastrophic forgetting and base model performance degradation.
>
> RE: Thank you for your valuable suggestion.
>
> To further assess potential catastrophic forgetting and general capability degradation, we evaluate three checkpoints on MMMU-Pro [1] (1,730 questions over 30 subjects) [1]:
> (i) the base Qwen2.5-VL-7B backbone,
> (ii) the model after Stage-I SFT on WritingPrompts,
> (iii) the model after Stage-II LDL-GRPO applied on top of SFT.
>
> Note that MMMU-Pro is used purely to probe whether our training harms the backbone’s general multimodal reasoning.
>
> **Summary results (overall performance):**
>
> | Metric | Base | SFT | LDL-GRPO |
> |---|---|---|---|
> | Macro Avg | 29.03 | 29.52 | 29.47 |
> | Micro Avg | 28.90 | 29.42 | 29.36 |
>
> These results show that the overall performance remains highly stable across stages, with only marginal changes (≤0.5%).  Overall, these results indicate that
> (1) Stage-I SFT does not cause noticeable catastrophic forgetting on MMMU-Pro;
> (2) LDL-GRPO further preserves general multimodal capability without introducing additional degradation.
>
> We will include the full three-way comparison in the revised version.
>
> [1] Mmmu-pro: A more robust multi-discipline multimodal understanding benchmark.
>
>
>
>
>
> >Q3.  confusing four numbers per task.
>
> RE: Thank you for the reviewer’s comment. We agree this can be confusing, although it is already specified in our experimental setup. All task details are provided in Appendix C.3. We set two sub-tasks for each dataset. For example, on A-OKVQA, we evaluate Task 5: A-OKVQA-MC and Task 6: A-OKVQA-RG: A-OKVQA-MC is a multiple-choice visual question answering task where the model selects the correct option based on the image and the question; A-OKVQA-RG (rationale generation) requires the model to produce an image-grounded, explainable rationale along with the answer.
>
> Meanwhile, under the same backbone, we report results for each sub-task using two external-judge settings: Raw win-rate (no length control) and LC win-rate (length-controlled). Therefore, for each backbone, each dataset yields 2 (sub-tasks) × 2 (Raw/LC) = 4 results.
>
>
> > Q4. What are the training datasets for Stage 1 and Stage 2?
>
> RE: Thank you for the suggestion.
>
> - In Stage 1 (SFT), we perform supervised fine-tuning on the training split of each dataset, which reflects a practical setting where users have access to limited in-domain labeled data.
>
> - In Stage 2 (RL), we assume no human preference annotations are available. Instead, we adopt a low-cost teacher-assisted approach: for each prompt, we query the black-box teacher once to obtain an anchor, and perform candidate sampling, ranking, and policy optimization locally (e.g., AnchorRank / LDL-GRPO).
>
> We will clarify this setup more explicitly in the revised version.

---

> > ### Author Rebuttal · Reviewer_56bu · 2026-04-02
> >
> > Thanks for the author's response. The additional objective metrics (exact match) demonstrated significant improvements on math tasks. Besides, the additional MMMU results also show no noticable forgetting. I'll therefore raise my score.

---

> > > ### Author Response · Authors · 2026-04-05
> > >
> > > Dear Reviewer 56bu,
> > >
> > > We are glad that our rebuttal addressed your concerns, and we are grateful for your thoughtful assessment 😊.
> > >
> > > Best regards,
> > >
> > > Authors

---

### Official Review · Reviewer_BTCx · 2026-03-15

**Soundness:** 2
**Presentation:** 3
**Significance:** 2
**Originality:** 2
**Overall Recommendation:** 3
**Confidence:** 4

**Summary:**

The paper proposes an RL-based distillation approach for aligning small language models using positive and unlabeled responses rather than explicit preference labels. The method is designed for on-premise deployment scenarios, where computational resources and access to large-scale annotation pipelines may be limited. By formulating the alignment problem as a positive–unlabeled (PU) learning task, the approach reduces the need for costly human annotations and frequent API calls to large teacher models. Experimental resultus suggest that models trained through this method is competitive with those obtained from standard baseline methods, such as PPO, GRPO.

**Compliance With Llm Reviewing Policy:**

Affirmed.

**Key Questions For Authors:**

See comments in the Strengths And Weaknesses section.

**Limitations:**

See comments in the Strengths And Weaknesses section.

**Strengths And Weaknesses:**

Strengths
- The target application scenario is interesting and practically relevant. The paper focuses on alignment under constraints such as privacy, cost, and latency, which commonly arise in on-premise deployment of small language models.
- The proposed method is clearly described, and the overall presentation is well organized and easy to follow. The paper provides a straightforward explanation of the motivation, formulation, and training pipeline.

Weaknesses
- The pipeline relies on self-ranking performed by the small student model, which may become a bottleneck. Errors in this ranking step could significantly affect the training signal. Misclassified positives could introduce biased reward signals.
- Following the previous comment, additional evaluation of the accuracy/performance at the ranking step should be considered to include. An additional experiment where a stronger model (e.g., the teacher model) performs the ranking would help quantify the impact of ranking errors and help to get a deeper understanding about the proposed pipeline.
- The ablation study is not clearly structured as a true ablation. It is not explicit which components of the method are being removed or modified, and the paper does not clearly summarize the insights gained from these experiments. A more systematic ablation analysis would help clarify the contribution of each component of the proposed method.
- The core idea appears closely related to prior work. In particular, it resembles the anchor-based RLAIF framework proposed in [1], and is also conceptually related to reinforcement learning from contrastive distillation (RLCD) introduced in [2]. The paper would benefit from a clearer discussion of the novelty and differences with respect to these methods.


[1] Li et al., Curriculum-RLAIF: Curriculum Alignment with Reinforcement Learning from AI Feedback, 2025 \
[2] Yang et al., RLCD: Reinforcement Learning from Contrastive Distillation for Language Model Alignment, 2024

---

> ### Author Rebuttal · Authors · 2026-03-31
>
> > Q1. self-ranking bottleneck.
>
> RE: Thank you for the comment. We understand the concern and then justify this as follows:
>
> - The teacher-provided anchor serves as a high-confidence reference, and ranking is performed via relative margins. This effectively cancels systematic scoring bias and brings the ranking back to a correct scale.
>
> -  Instead of hard pairwise comparisons, we use listwise supervision over a candidate group, which diffuses errors and reduces sensitivity to noisy rankings. Theorem 3.2 further shows that the induced distribution concentrates on near-optimal candidates with bounded deviation.
>
> - From Table 1 and the ablation study, methods relying on scalar self-reward (e.g., Self-PPO) exhibit instability, whereas our method yields consistent and stable improvements.
>
> - By the way, we report the Spearman correlation (SC)  between our method’s (Qwen2.5 vl based) ranking and the teacher (GPT4o) model’s ranking on 2 datasets. Note that the SC ranges from -1 to +1: +1 indicates a perfect positive correlation,  and -1 indicates a perfect negative correlation.
>
> | Method | A-OKVQA-Rationale | Counting Probability |
> |---|---|---|
> | SFT MODEL | 0.472 | 0.528 |
> | SELF-GRPO | 0.561 | 0.604 |
> | Ours | **0.628** | **0.700** |
>
> These results show that:
>
> (i) the anchor brings the ranking back to a correct scale (0.472 → 0.561), and
>
> (ii) LDL further refines and stabilizes the preference structure (0.561 → 0.628 / 0.700).
>
> Overall,  our method alleviates the bottleneck of small models while operating under lower cost.
>
> Thanks again! We will highlight this in the version.
>
>
>
>
>
> > Q2. ranking  performance with teacher.
>
> RE: Thanks. To address the concern, we report the SC between our method’s  ranking and the teacher (GPT4o) model’s ranking on A-OKVQA-Rationale and Counting Probability datasets.
>
> | Method | A-OKVQA-Rationale | Counting Probability |
> |---|---|---|
> | Ours | **0.628** | **0.700** |
>
> The results show that our ranking is close to the teacher’s ranking. Moreover, this observation is consistent with Theorem 3.2, which guarantees that the induced listwise distribution concentrates on near-optimal candidates, leading to improved rank consistency with the teacher preference.
>
> Thanks again! We will update this in the revision.
>
> > Q3. ablation study is not clearly.
>
> RE: Thank you for the comment.  In the main text, we have already explained the two components of our method, which can be summarized as (C1) anchor-based calibration and (C2) listwise LDL supervision. In Finding 1 (Line 322), we state that (C1)  consistently outperform non-anchor methods. In Table 2, both Anchor-GRPO and LDL-GRPO are anchor-based; we further observe that adding C2 yields additional improvements. To reduce confusion, we re-list the ablation results as below.
>
> We re-list the results below for clarity. Each cell reports two scores  under  Raw / LC  (details see lin 297Implementation).
> - (A0) Self-GRPO;
> - (A1) Anchor-GRPO = A0 + C1;
> - (A2) LDL-GRPO = A1 + C2.
>
> | Domain | Task | Backbone | SFT | A0 | A1 | A2(ours) |
> |---|---|---|---|---|---|---|
> | Unimodal | CountProb | Qwen2.5-7B | 0.533 / 0.487 | 0.548 / 0.515 | 0.558 / 0.538 | **0.578 / 0.573** |
> | Multimodal | A-OKVQA-MC | Qwen2.5-VL-7B | 0.131 / 0.136 | 0.126 / 0.133 | 0.146 / 0.156 | **0.236 / 0.231** |
>
> The results show consistent monotonic improvements from A0→A1→A2, indicating that:
>
> (i) Anchor-based calibration  stabilizes preference induction;
>
> (ii) LDL/listwise supervision  further improves performance via more robust and informative preference signals.
>
> We will update the ablation table clearly  in the revision.
>
>
> > Q4. discussion with RLCD and RLAIF
>
> RE: Thanks for the suggestion. We agree that clarifying the relation to RLCD (1) and RLAIF (2) would improve the paper.   We also emphasize that our method is different from these two works, and the discuss are summarized  as below
>
> > diss with 1:
>
> - RLCD relies on self-RLHF, where the model performs self-ranking that can be inaccurate;
> Ours introduces an anchor-based alignment mechanism, which mitigates the inaccuracy of self-ranking by providing a stable reference.
> - RLCD requires a reward model , which may not be available  in real-world enterprise settings. Ours does not require any reward model.
> - RLCD adopts pairwise supervision; Ours uses listwise supervision over K candidates, where the model assigns a preference distribution over a group of responses, reducing the impact of inaccurate self-ranking.
>
>
> > diss with 2:
>
> -  RLAIF relies on a reward model, which may not be available  in real-world enterprise settings, especially for domain-specific tasks.
> - RLAIF requires repeated API calls to evaluate each candidate response, leading to substantial computational and financial overhead.
> - In contrast, our method enables alignment transfer under  low-cost, using only one teacher call per prompt without per-candidate scoring.
>
> We will update the discussion  in the revision.

---

> > ### Author Rebuttal · Reviewer_BTCx · 2026-04-02
> >
> > Thanks to the authors for the additional clarification and experimental results. The reported Spearman correlation (SC) scores indicate an improvement of the proposed method over the SFT model. However, an SC value in the range of 0.6–0.7 reflects only a moderate correlation, which may not be sufficient to claim strong alignment with the teacher model’s predictions. Furthermore, the evaluation is limited to only two datasets, which restricts the generalizability of the findings. Additional experiments on a broader set of benchmarks would help strengthen the claim that the performance of the student model is highly correlated with that of the teacher model.

---

> > > ### Author Response · Authors · 2026-04-03
> > >
> > > Q5. Spearman correlation insufficient, and additional experiments are needed.
> > >
> > >
> > >
> > > RE: Thank you for the comment.  Due to space limitations, our previous response results on a subset of datasets.
> > > In this response, we not only report
> > > (1) ranking results on **six datasets** using two correlation metrics；
> > > (2) but also provide a **statistical significance analysis** of the ranking differences between our method and the teacher.
> > >
> > > These results indicate that:
> > > - In most cases, the rankings produced by our anchor-based method are **not statistically significantly different** from the teacher’s rankings;
> > > - The **anchor** helps bring the ranking back to the correct scale; and
> > > -  **LDL** further refines and stabilizes the learned preference structure.
> > >
> > >
> > > Detailed results are provided below.
> > >
> > >
> > > > ## 1） Statistical analysis.
> > >
> > >
> > >
> > > We performed multiple t-tests across different datasets and counted how many times our ranking scores and the teacher model’s scores show **no statistically significant difference**. Specifically, for each dataset containing \(n\) samples (prompts), we obtain a \(K\)-dimensional score vector for each prompt from: (i) our method’s scores over the \(K\) sampled candidate responses; and (ii) the teacher model’s scores over the same \(K\) candidate responses. We then conduct a two-sided t-test between these two score vectors (ours vs. teacher) for each prompt. This results in \(n\) t-tests per dataset. Finally, we report, across multiple datasets, the number of t-tests that are not statistically significant ( \(p > 0.05\)).
> > >
> > >
> > >
> > >
> > >
> > >
> > > | Dataset              | Prompts number | Significant number (p ≤ 0.05) | Non-significant number (p > 0.05) | Non-significant Ratio (%) |
> > > | -------------------- | -------------- | ----------------------------- | --------------------------------- | ------------------------- |
> > > | CFCW                 | 2636           | 419                           | 2217                              | 84.1                      |
> > > | PBFG                 | 2458           | 380                           | 2078                              | 84.5                      |
> > > | Counting Probability | 498            | 114                           | 384                               | 77.1                      |
> > > | Geometry             | 540            | 93                            | 447                               | 82.8                      |
> > > | A-OKVQA-MC           | 1335           | 247                           | 1088                              | 81.5                      |
> > > | A-OKVQA-Rationale    | 7280           | 1732                          | 5548                              | 76.2                      |
> > > | Total                | 14747          | 2985                          | 11762                             | 79.8                      |
> > >
> > >
> > > The per-prompt two-sided t-tests show that, across six datasets, most prompts yield **no statistically significant difference** between our score vectors and the teacher’s (overall **79.8%** non-significant). This pattern is consistent across unimodal and multimodal tasks (**76.2%–84.5%**). Even on harder tasks such as **Counting Probability** and **A-OKVQA-Rationale**, the non-significant ratios remain high (**77.1%** and **76.2%**, respectively), suggesting our anchor-based optimization produces score patterns largely consistent with the teacher at the prompt level.
> > >
> > >
> > >
> > >
> > >
> > >
> > >
> > > > ## 2） Six datasets results.
> > >
> > > ## Spearman correlation
> > >
> > > | Method    | CFCW  | PBFG  | Counting Probability | Geometry | A-OKVQA-MC | A-OKVQA-Rationale | Avg.  |
> > > | --------- | ----- | ----- | -------------------- | -------- | ---------- | ----------------- | ----- |
> > > | SFT MODEL | 0.574 | 0.603 | 0.528                | 0.556    | 0.491      | 0.472             | 0.537 |
> > > | SELF-GRPO | 0.662 | 0.694 | 0.604                | 0.646    | 0.583      | 0.561             | 0.625 |
> > > | Ours      | 0.803 | 0.826 | 0.700                | 0.768    | 0.721      | 0.628             | 0.741 |
> > >
> > >
> > >
> > >
> > > ##  Pearson correlation
> > > | Method    | CFCW  | PBFG  | Counting Probability | Geometry | A-OKVQA-MC | A-OKVQA-Rationale | Avg.  |
> > > | --------- | ----- | ----- | -------------------- | -------- | ---------- | ----------------- | ----- |
> > > | SFT MODEL | 0.635 | 0.662 | 0.585                | 0.624    | 0.563      | 0.512             | 0.597 |
> > > | SELF-GRPO | 0.742 | 0.764 | 0.674                | 0.724    | 0.652      | 0.603             | 0.693 |
> > > | Ours      | 0.872 | 0.889 | 0.775                | 0.833    | 0.804      | 0.726             | 0.817 |
> > >
> > > Across six datasets, our method achieves the highest teacher-ranking agreement under both Spearman and Pearson correlations (Avg. Spearman: 0.537→0.625→0.741; Avg. Pearson: 0.597→0.693→0.817 for SFT→Self-GRPO→Ours), with notably strong gains on the harder **Counting Probability** and **A-OKVQA-Rationale** tasks, supporting better alignment with the teacher’s preference structure.
> > >
> > >
> > >
> > >
> > >
> > >
> > >
> > >
> > >
> > > Thanks again! We will update this in the revision.

---

### Decision · Program_Chairs · 2026-04-30

**Decision:**

Accept (regular)

**Comment:**

This paper proposes a novel method that avoids extensive data collection, and the setting is well-motivated. Empirical results show consistent performance improvements.

There were quire a few discussion points and extra results during the discussion. Please take them into account for the camera-ready version.